# Impact of Multi-Professional Intervention on Health-Related Physical Fitness and Biomarkers in Overweight COVID-19 Survivors for 8 and 16 Weeks: A Non-Randomized Clinical Trial

**DOI:** 10.3390/healthcare12202034

**Published:** 2024-10-14

**Authors:** Marielle Priscila de Paula Silva-Lalucci, Déborah Cristina de Souza Marques, Joed Jacinto Ryal, Marilene Ghiraldi de Souza Marques, Victor Augusto Santos Perli, Ana Flávia Sordi, Solange Marta Franzoi de Moraes, Pablo Valdés-Badilla, Leonardo Vidal Andreato, Braulio Henrique Magnani Branco

**Affiliations:** 1Interdisciplinary Laboratory of Intervention in Health Promotion, Cesumar Institute of Science, Maringá 87050-390, Brazil; mariellepriscila@gmail.com (M.P.d.P.S.-L.); marques.deborah@hotmail.com (D.C.d.S.M.); joed.ryal@unicesumar.edu.br (J.J.R.); marileneghiraldi@gmail.com (M.G.d.S.M.); victoraperli@gmail.com (V.A.S.P.); anaflaviasordi@gmail.com (A.F.S.); vidal.leo@hotmail.com (L.V.A.); 2Graduate Program in Health Promotion, Cesumar University, Maringá 87050-390, Brazil; 3Graduate Program of Human Physiology, State University of Maringá, Maringá 87020-900, Brazil; smfmoraes@uem.br; 4Department of Physical Activity Sciences, Faculty of Education Sciences, Universidad Católica del Maule, Talca 3530000, Chile; valdesbadilla@gmail.com; 5Sports Coach Career, School of Education, Universidad Viña del Mar, Vina del Mar 2520000, Chile; 6Physical Education Course, State University of Amazonas, Barcelos 69700-000, Brazil

**Keywords:** coronavirus, interdisciplinary study, health promotion, multi-professional team, rehabilitation research

## Abstract

Background/objectives: Considering the diverse symptomatology of COVID-19—ranging from mild to severe cases—multi-professional interventions are crucial for enhancing physical recovery, nutritional status, and mental health outcomes in affected patients. Thus, this study aimed to investigate the effects of such an intervention on health-related physical fitness and biomarkers in overweight COVID-19 survivors with varying degrees of symptom severity after 8 weeks and 16 weeks. Methods: This non-randomized clinical trial included 59 overweight COVID-19 survivors (32 males and 27 females) divided into three groups: mild (*n =* 31), moderate (*n =* 13), and severe/critical (*n =* 15). The participants underwent a multi-professional program and were assessed for anthropometric and body composition (primary outcome), as well as physical fitness and biochemical markers (secondary outcome) 8 and 16 weeks before the intervention. Results: After 8 weeks, time effects were observed for the maximum isometric handgrip strength (*p* < 0.001), maximum isometric lumbar-traction strength (*p* = 0.01), flexibility (*p* < 0.001), abdominal strength–endurance (*p* < 0.001), the sit-and-stand test (*p* < 0.001), maximum oxygen consumption (*p* < 0.001), and distance covered in the 6 min walk test (*p* < 0.001). Additionally, time effects were also observed for fat mass (*p* = 0.03), body fat percentage (*p* = 0.02), abdominal circumference (*p* = 0.01), total cholesterol (*p* < 0.001), low-density lipoproteins (*p* < 0.001), and glycated hemoglobin (*p* < 0.001), with lower values after multi-professional interventions. After 16 weeks, the systolic and diastolic blood pressure showed significant reductions independently of the intervention group (*p* < 0.001). Conclusion: These findings suggest that multi-professional interventions can provide substantial benefits for post-COVID-19 patients, regardless of the severity of their initial symptoms.

## 1. Introduction

Coronavirus-19 (COVID-19) is an acute respiratory infection caused by the SARS-CoV-2 coronavirus, which is potentially severe, highly transmissible, and globally distributed [1]. Some COVID-19 survivors have presented sequelae of a respiratory order; physical deconditioning, with a loss of musculoskeletal mass and reduced muscle strength and endurance; a decreased cardiorespiratory capacity and reduced quality of life; and emotional problems, among others [1]. Unlike the signs and symptoms of COVID-19, the complications of acute post-COVID-19 are not fully understood [2]. For some people, complications may affect multiple organs and persist for months, regardless of the severity of the disease at onset. However, persons with a higher degree of COVID-19 involvement have a higher risk of death within 12 months of illness [3].

The severity of the disease depends on the intensity of the immune response triggered by the virus. Therefore, long-term signs and symptoms are subject to the extent and severity of the viral infection, the organs affected, and the so-called “cytokine storm” during the acute phase of COVID-19 [4]. The acute post-COVID-19 syndrome or post-COVID-19 state includes fatigue, dyspnea, chest pain, a loss of taste and/or smell, cognitive changes, and arthralgias, which are factors that affect work and daily functioning for a long time due to COVID-19 [5,6]. Thus, rehabilitation strategies for COVID-19 survivors are indispensable to combat a condition with different sequelae, minimize damage, and significantly impact the population, the health of persons, and the economy [5,6,7,8].

In addition, an intervention applied to COVID-19 survivors must consider that it is well established in the scientific literature that obesity is a risk factor for severe COVID-19, with a dose–response relationship between a higher body mass index (BMI) and worsening of the disease [6,7,9]. Obesity is a low-grade pro-inflammatory disease correlated with the increased prevalence of other chronic non-communicable diseases (NCDs), such as type 2 diabetes mellitus, hypertension, dyslipidemia, and metabolic syndrome, among other pathologies [10]. The scientific literature showed a higher prevalence of hospitalization in obese patients with COVID-19. Rottoli et al. [11] carried out a study in a hospital in Bologna, Italy; among the obese individuals, 51.9% had respiratory failure, 36.4% were admitted to the intensive care unit, 25% required mechanical ventilation, and 29.8% died within 30 days of the onset of symptoms. Gao et al. [12] investigated the association between obesity and the severity of COVID-19 in three Chinese hospitals, and found that obese individuals had more extended hospital stays and progressed to the severe form of the disease.

In addition, biochemical markers could be altered in COVID-19 survivors to different degrees, like fasting glucose, glycated hemoglobin, triglycerides, C-reactive protein, and other biomarkers [5]; however, multi-professional interventions may promote benefits in the biochemical responses for health for 8 weeks of intervention [5]. Therefore, the application of interventions aimed at patient recovery by providing physical activity, healthy eating, and psychoeducation could reduce the sequelae of the disease and the complications resulting from the post-COVID syndrome in overweight persons [5,6,7,8].

In this sense, interventions with physical exercise, dietary re-education, and psychoeducation in post-COVID-19 patients after 8 weeks showed promising responses of multi-professional interventions with a significant reduction in the C-reactive protein, a significant increase in serum albumin, and a significant improvement in the sit-to-stand test in those who were hospitalized [5]. Additionally, other studies also explored the function of physical exercise as a possible approach to mitigate the harmful impacts of COVID-19. However, they have not yet considered the various multi-professional elements related to public health promotion strategies and the severity of COVID-19 in different symptomatologies (mild, moderate, and severe/critical cases) [13,14]. Additionally, the multi-professional intervention for patients surviving COVID-19 for more than 12 weeks, considering the specific symptoms (mild, moderate, and severe/critical) [13,14], is still unknown to the present study’s authors. Consequently, considering the symptoms between the groups may be relevant to promoting assertive interventions and recovery prognoses.

Thus, this study aimed to investigate the effects of a multi-professional intervention on health-related physical fitness and biochemical parameters in overweight and obese COVID-19 survivors (in different degrees of severity) for 8 and 16 weeks. Drawing from earlier research [5,13,14], the authors of this study suggest, as a primary outcome, that an 8-week multidisciplinary intervention model can enhance anthropometric and body composition parameters. As a secondary outcome, it can improve physical fitness and metabolic parameters, irrespective of the disease’s symptomatology. Furthermore, extending the intervention to 16 weeks may yield more consistent results than the 8-week program.

## 2. Materials and Methods

### 2.1. Experimental Approach to the Problem

This study is an uncontrolled parallel-group non-randomized clinical trial, with repeated measures over 8 and 16 weeks, carried out from January to October 2022, following Consolidated Standards of Reporting Trials (CONSORT) [15]. The experimental groups (mild COVID-19 group, moderate COVID-19 group, and severe/critical COVID-19 group) underwent a multi-professional program consisting of physical exercises (muscle strength and aerobic exercises, i.e., concurrent training), nutritional intervention, and psychoeducation. Participants were assessed at the beginning of the study (pre-intervention), after 8 weeks (post-8 weeks), and after 16 weeks of intervention (post-16 weeks). Those interested contacted the multi-professional team from the Interdisciplinary Health Promotion Intervention Laboratory in the Cesumar University facilities.

The assessments included anthropometry and body composition (primary outcome), flexibility, maximal isometric strength tests, dynamic muscle strength–endurance, cardiorespiratory fitness, and biochemical parameters (secondary outcome). All assessments were conducted in the morning (between 7:00 and 11:30 h) and in the exact location (laboratory, with the control of variables, temperature, and investigators that applied the procedures). The patients did not present pain before the assessments or during the training sessions without presenting musculoskeletal and/or cardiorespiratory injuries during the intervention. The present study was approved by the Local Research Ethics Committee (protocol n° 4.546.726/2021) and followed the Declaration of Helsinki. This study was registered in the Brazilian Clinical Trials Registry Platform (REBEC) under RBR-4mxg57b. All subjects were informed about the purposes of the study and signed an informed consent form.

Baseline measurements were taken over two days. First, the subjects underwent a clinical assessment by a pulmonologist and an intensive care physician, consisting of the participant’s clinical history (history of surgery, pre-existing chronic non-communicable diseases, continuous use of medication, main signs and symptoms showing possible sequelae of COVID-19, and type and length of stay in hospital (ward/room or intensive care unit)), anthropometric and body composition assessment, and blood collection for biochemical analyses. Figure 1 shows the methodological design of the present study.

On the second day, the following data were collected: (i) blood pressure [BP—systolic blood pressure (SBP) and diastolic blood pressure (DBP)] after 5 min of rest, according to the VIII Guideline on Arterial Hypertension [15]; (ii) measurement of heart rate (HR) and peripherical oxygen saturation (%SpO_2_), both at rest; (iii) posterior chain flexibility test on the Wells bench (sit and reach test); (iv) maximal isometric strength tests with specific dynamometers; (v) sit-up test; (vi) 30 s chair–stand test; (vii) push-up (adapted); and (viii) 6 min walk test (6MWT). The physical fitness tests were performed following the recommendations published earlier by Sordi et al. [5] in a clinical trial with post-COVID-19 patients. After the 6MWT, the following variables were collected: BP, HR, and %SpO_2_. All tests are described in the sections below. Moreover, all evaluators had experience carrying out measurements and had already participated in other studies measuring the same indicators presented in the present study, with an intra-evaluator reproducibility greater than 0.95 with data from our research laboratory. After the clinical assessment, the self-reported signs and symptoms were considered for the non-randomized allocation of participants in the experimental COVID-19 groups according to the “Clinical Management of COVID-19: Living Guidance” [16]. The clinical manifestations of COVID-19 are divided according to their severity: mild (no evidence of viral pneumonia or hypoxia); moderate (clinical signs of pneumonia, but no signs of severe pneumonia); severe (clinical signs of pneumonia plus one of the following: respiratory rate >30 breaths/min, severe respiratory distress, or SatO_2_ < 90%); and critical (acute respiratory distress syndrome, sepsis, septic shock, or acute thrombosis) [16]. Over the 16 weeks, 16 participants dropped out of the study for different reasons. Figure 2 presents the flowchart of the present study’s participants based on the CONSORT Guidelines [17].

### 2.2. Study Participants

Participants were recruited through the Maringá, Paraná, Brazil, Municipal Health Department, Municipal Hospital, and TV, radio, and social media advertising. One hundred and forty-six volunteers of both sexes were invited to take part in the study according to the following inclusion criteria: (i) aged between 19 and 65 years; (ii) BMI ≥ 25.0 kg/m^2^; (iii) positive diagnosis confirmed by RT-PCR (reverse transcriptase polymerase chain reaction) for COVID-19; (iv) medical authorization to participate in this study; (v) first dose of the COVID-19 vaccine; (vi) being available to participate in multi-professional interventions twice a week for sixteen weeks; and (vii) having contracted COVID-19 between 2 January 2021 and 22 September 2021. Exclusion criteria included the following: (i) debilitating neurological diseases (i.e., Alzheimer’s or Parkinson’s); (ii) contraindications for physical exercise; and (iii) pregnancy. Following Jensen et al. [18], a minimum of 15 participants per group would be sufficient to achieve a statistical power of 80% with an alpha error of 5%. One hundred and forty-six volunteers randomized in three experimental groups were eligible, but seventy-one were excluded because they did not meet the inclusion criteria. In total, seventy-five volunteers were accepted to participate in the program and were allocated according to their COVID-19 symptoms: fifteen volunteers were allocated to the mild COVID-19 group, eighteen volunteers to the moderate COVID-19 group, and sixteen volunteers to the COVID-19 severe/critical group. At the end (after 16 weeks), 59 volunteers were analyzed: mild COVID-19 (*n =* 31), moderate COVID-19 (*n =* 13), and severe/critical COVID-19 (*n =* 15) (Figure 1).

### 2.3. Medical Clearance

First, all patients underwent a medical consultation with an intensive care physician/thoracic surgeon. The physician reviewed the patient’s medical records, blood tests, computed tomography scans, type of hospitalization, use of ventilators, %SatO_2_, blood pressure, and cardiological exams (to assess for possible myocarditis or pericarditis) and performed pulmonary and cardiac auscultation. Additionally, the physician conducted an anamnesis to identify any prolonged symptoms of COVID-19, such as fatigue, dyspnea, or other lingering effects. Based on this comprehensive evaluation, the physician determined whether the patient could engage in physical activity. Patients deemed unfit for physical activity were referred for cardiopulmonary or neurological physiotherapy rehabilitation, as previously described by Lemos et al. [7] in an earlier study conducted by the same research group.

### 2.4. Procedures

The participants’ stature was measured using a stadiometer attached to a scale with a capacity of 2.2 m and an accuracy of 0.1 cm (Welmy R-110^®^, Santa Bárbara D’ Oeste, São Paulo, Brazil). Abdominal circumference was measured using a tape measure (model T87-2^®^, Florianopolis, Santa Catarina, Brazil), with a measuring capacity of 2 m and precision of 0.1 cm, following the specifications proposed by Heyward [19]. The participant’s body composition was measured using tetrapolar bioimpedance (InBody 570^®^, Bio space Co., Ltd., Seoul, Republic of Korea), with a capacity of 250 kg and an accuracy of 100 g, according to the manufacturer’s instructions, following recommendations to improve validity [20]. All participants were previously instructed on the recommendations. The following parameters were measured: BMI (kg/m^2^), lean mass (kg), fat mass (kg), body fat percentage (%), and skeletal muscle mass (kg) [21].

### 2.5. Biochemical Analyses

The blood collection procedures followed the guidelines of the Clinical and Laboratory Standards Institute [22]. Participants were previously instructed on how to prepare for the collections at the Clinical Analysis Laboratory of the Cesumar University facilities, and after collection, participants were instructed to press on the puncture site to avoid bruising. The collected blood samples were dispensed into Vacuplast^®^ collection tubes containing stacking gel with an activator and tubes with the anticoagulant ethylene diamine tetra acetic acid (EDTA) K2. Subsequently, to obtain serum, the samples containing an activator were centrifuged in a Centrilab^®^ analog centrifuge at 3.500 rpm (relative centrifugal force) for 15 min at room temperature. The following laboratory tests were analyzed: glycemic control [glycated hemoglobin: (HbA1c)]; lipid profile [total cholesterol, high-density lipoprotein (HDL-c), low-density lipoprotein (LDL-c), and triglycerides (TGL)], liver enzymes [alanine aminotransferase (ALT), aspartate aminotransferase (AST), gamma-glutamyl transferase (GGT), alkaline phosphatase (ALP), and albumin], C-reactive protein (CRP), markers of renal function (creatinine and urea), electrolytes (magnesium, total calcium, and phosphorus), and markers of pancreatic function (amylase and lipase). The analyses used the Gold Análise Diagnóstica kits (Belo Horizonte, Minas Gerais, Brazil) in the automatic biochemical and turbidimetric analyzer device URIT 8021^®^ from MHLab. All analyses were performed in triplicate. The Finecare^®^ FIA Meter Plus analyzer from WONDFO was used for HbA1c.

### 2.6. Physical Fitness

#### 2.6.1. Health-Related Physical Fitness Tests

The chosen physical tests to evaluate the outcomes of the COVID-19 survivors follow the order: (i) sit and reach test; (ii) maximal isometric handgrip strength (MIHS), maximal isometric lumbar-traction strength (MILTS); (iii) sit-up test for abdominal strength resistance; (iv) 30 s chair–stand test for lower limbs; (v) push-up test for upper limbs; and (vi) 6MWT [1]. The participants were instructed about the procedures for all physical tests, and the researchers respected the remaining 5–10 min between the tests (in each test, the participants were familiarized with performing the tests), conforming to the orientation of Sordi et al. [5]. Furthermore, the choice of physical tests was based on promoting the assessment of physical fitness test parameters in places with low resources, clinics, public hospitals, gyms, and others.

##### Flexibility Assessment

The sit and reach test was employed to evaluate the flexibility of the posterior chain using the Wells Bench. Participants were instructed according to previously described procedures [23]. The test was repeated three times, with a 60 s interval between attempts. The highest value obtained was recorded and expressed in cm.

##### Maximal Isometric Strength Tests (MIHS and MILTS)

To assess MIHS, a TKK 5101 dynamometer (Takei Physical Fitness Test^®^, Tokyo, Japan) with a capacity of 100 kg was used. MILTS was evaluated using a Takei dynamometer (Takei Physical Fitness Test^®^, Back Strength Dynamometer, type 2, Japan) with a capacity of 300 kg. According to previous recommendations, three trials were performed for both tests, lasting 3–5 s with a 1 min rest between trials [24]. The highest value was recorded in kgf.

##### Dynamic Muscle Strength–Endurance Assessment

To assess dynamic muscle, sit-up, 30 s chair–stand, and push-up tests for upper limbs were performed according to the procedures described in previous studies [25,26]. For the sit-up and push-up tests, the maximum number of repetitions achieved in 60 s was recorded, and for the 30 s chair–stand test, the muscular endurance of the lower limbs was evaluated from the maximum number of repetitions performed.

##### Cardiorespiratory Fitness Test

The 6MWT was applied to verify the cardiorespiratory fitness of the present study participants. The 6MWT test was performed per American Thoracic Society guidelines [27]. Volunteers were instructed to walk as fast as possible to achieve the greatest distance at the end of 6 min. [27]. Before and after the 6MWT, distance, heart rate, and systolic and diastolic pressure were collected. The peak oxygen consumption (VO_2peak_) was calculated using a previous study [28].

### 2.7. Multi-Professional Intervention

#### 2.7.1. Food Re-Education Protocol

Nutritionists held nutritional orientation meetings in groups, with theoretical and practical activities to promote the change in eating behavior. The classes, inspired by the Food Guide for the Brazilian Population [29], were adapted to the new scenario experienced by COVID-19. The aim was to educate participants with dynamic classes on the benefits of healthy eating for health and how to deal with the risks associated with chronic NCDs and long-term symptoms of COVID-19 [5]. The meetings took place once a week for approximately 45 min for 16 consecutive weeks. In addition to dynamic classes, printed materials on the theme were prepared to be delivered. The interventions addressed the following themes, conforming to Table 1.

#### 2.7.2. Psychoeducation Protocol

The psychologists held psychoeducation meetings to prevent and treat mental illness with an educational character. The protocol is applied using concepts and information from psychology, based on cognitive–behavioral theory (CBT), combined with other areas so that the individual can gain a broad understanding of their situation and other mental illnesses related to their condition, as well as focus on prevention and health promotion [8,30,31]. The meetings took place in a group once a week, for approximately 45 min, for 16 consecutive weeks. The interventions were based on content presentations, conversation circles, and dynamics. The psychoeducational interventions were adapted to the participants’ reality according to the feedback provided during the interventions. In addition to the dynamic lessons, printed materials on the subject were prepared for delivery. The psychoeducation interventions addressed the following themes, conforming to Table 2.

To reinforce the interventions applied and disseminate support material to participants, family members, and the community, we used printed and digital information papers delivered and sent after each intervention [30,31,32,33]. The interventions aimed to provide knowledge and enable changes about the psychological consequences of the COVID-19 pandemic, in addition to providing guidance on the essential themes of our century and helping participants to become more aware of the mental disorders caused by the contagion process and the COVID-19 pandemic.

#### 2.7.3. Physical Exercise Protocol

The physical exercise intervention sessions were held at the Cesumar University facilities twice weekly, lasting 60 min each, with alternating training programs A and B on different days. The training protocol consisted of cardiorespiratory and muscle strength exercises (concurrent training) to increase muscle strength and, if necessary, motor coordination and balance. The concurrent training plan consisted of 4 weeks of anatomical adaptation with low volume and intensity, that is, 3 sets of 15 repetitions, with no aerobic exercise at the end of the session, and the remaining weeks of physical exercise (12 more in total) had a gradual progression of volume and intensity (via the classic linear periodization); in other words, the loads used were readjusted over the weeks, as well as the number of sets and repetitions. In weeks 5 to 8, 3 sets of 12 repetitions were performed; in weeks 9 to 12, the training sessions consisted of 4 sets of 12 repetitions; and finally, in weeks 13 to 16, 4 sets of 20 repetitions were performed. The rating of perceived exertion (RPE) for resistance training was used to regulate the intensity of the training sessions [34], where 0 = rest and 10 = maximal effort, with values measured in arbitrary units (a.u.). The study participants were familiarized with the scale. The RPE scale remained in the strength training laboratory, allowing patients to monitor the intensity of their sessions. During the first four weeks, the session intensity ranged from 1 a.u. to 4 a.u.; in weeks 5 to 8, the intensity gradually increased from 4 a.u. to 6 a.u.; between the 8th and 12th weeks, the intensity ranged from 6 a.u. to 7 a.u.; and from the 13th to the 16th week, the intensity fluctuated between 5 a.u. and 8 a.u. This protocol was based on/adapted from the interventions with post-COVID-19 patients conducted earlier by Sordi et al. [5]. About aerobic exercise, 1 set of 5 and 1 set of 10 min was performed on weeks 5 to 8; 1 set of 10 and 1 set of 15 min was performed on weeks 9 to 12; and finally, on weeks 13 to 16, 1 set of 10 and 1 set of 20 min was performed. The training was carried out with resistance exercises focused on large muscle groups and cardiorespiratory fitness, which were performed on a treadmill, vertical/horizontal bicycle, or rowing ergometer, according to the preference and physical condition of the patients. The rest interval between exercises varies for each participant, depending on their physical condition. The rest intervals were individualized based on each participant’s readiness to perform the exercise. As a result, rest periods were self-controlled and tailored to each participant. After each exercise and set, participants were asked if they were ready to continue, with rest times varying between 45 and 90 s. Due to restrictions imposed by the COVID-19 pandemic and/or the risk of infections, it was decided to conduct physical activities only twice a week to minimize the exposure of participants and study staff to potential risks of contracting SARS-CoV-2.

Consequently, the exercise sessions were designed at low volume and frequency and performed twice weekly. This approach has shown therapeutic benefits across various chronic diseases, as evidenced by Pedersen and Saltin’s review [35]. It has been effective in improving symptoms, functional capacity, and quality of life in patients with psychiatric, neurological, metabolic, cardiovascular, pulmonary, and musculoskeletal disorders and cancer. In addition, ten percent of severe/critical COVID-19 patients had their warm-up routines adapted to include coordination and balance activities. The activities performed were standing on one leg, swinging the legs, raising the legs backward, raising the arms, standing on tiptoes, walking in a straight line, sitting down and standing up from a chair, jumping over objects, using a Pilates ball, walking on a slackline, and standing on a Bosu ball—after that, they performed the physical exercises conforming to Table 3. Table 3 presents the training program performed by the experimental groups during the 16 weeks of multi-professional intervention.

### 2.8. Statistical Analysis

The statistical analyses were conducted using the SPSS 24 software version (IBM, Armonk, NY, USA). Data are presented as the mean ± standard deviation (SD) and 95% confidence interval (95% CI). Data normality was tested using the skewness–kurtosis test, considering values from 2 to −2 to indicate a need for parametric statistical analyses. The main effect and interaction between groups and time were performed via a two-way mixed-measure ANOVA (for repeated measures). Bonferroni’s post hoc test was used when a significant difference was found. The significance level established for all tests was *p* < 0.05. The partial eta square (η^2^) was calculated according to the classification by Richardson [36] using the following interpretation scale: 0.0099 [small], 0.0588 [moderate], and 0.1379 [large]. Cohen’s (d) was also calculated for effect size using the following rating: 0.20 [small], 0.80 [moderate], and >0.80 [large] [37].

## 3. Results

Table 4 shows the clinical characteristics of the participants stratified by COVID-19 symptoms. No differences were observed for age, sex, BMI, resting heart rate, SBP, BPD, and %SpO_2_ (*p* > 0.05 for all comparisons). Regarding the persistent symptoms self-reported by COVID-19 patients, a memory deficit (mild: 71.0%; moderate: 69.2%; severe/critical: 60.0%), fatigue (mild: 41.9%; moderate: 53.8%; severe/critical: 46.7%), and muscle pain (mild: 32.3%; moderate: 46.2%; severe/critical: 53.3%) were more prevalent. However, there was no significant difference between the groups for a memory deficit, fatigue, and muscle pain (*p* > 0.05 for all comparisons). Regarding the participants’ medical history, a difference was only observed for heart disease (mild: 19.4%; moderate: 0%; severe/critical: 40.0%; with higher values for the severe/critical group, *p* = 0.03). In addition, no differences were detected for the other clinical characteristics (self-reported post-COVID-19 symptoms and physical activity: *p* > 0.05). In addition to the results presented in the article, Appendix A were created to present data by sex: anthropometry, body composition, physical tests and biochemical variables for men and women, without considering the symptoms of COVID-19 and tables making the distinction by sex, considering symptoms of COVID-19). 

### 3.1. Anthropometric and Body Composition Measurements

A time effect was observed for the abdominal circumference (F = 5.08; *p* = 0.01; η^2^p = 0.08—moderate effect), with a significant reduction after 8 weeks (*p* = 0.01) and no significant changes after 16 weeks (*p* > 0.05). No time effects were observed for the weight, BMI, fat-free mass, musculoskeletal mass, fat mass, and body fat percentage (*p* > 0.05). A group effect was observed for fat mass (F = 3.91; *p* = 0.03; η^2^p = 0.12—moderate effect), with significantly higher values for the severe/critical group compared to the mild group (*p* = 0.02), and for the body fat percentage (F = 4.54; *p* = 0.02; η^2^p = 0.14—large effect), with significantly higher values for the severe group when compared to the mild group (*p* = 0.04). No group effects were observed for weight, BMI, and abdominal circumference (*p* > 0.05). None of the variables showed an interaction effect between the group and time measurements (*p* > 0.05).

Table 5 presents the anthropometry and body composition pre-test after 8 and 16 weeks of intervention.

### 3.2. Health-Related Physical Fitness Tests Responses

A time effect was observed for all tests: MIHS of the right side (F = 14.00; *p* < 0.001; η^2^p = 0.22—large effect), with a significant increase after 8 weeks (*p* = 0.01); MIHS of the left side (F = 10.34; *p* < 0.001; η^2^p = 0.17—large effect), with a significant increase after 8 weeks (*p* = 0.01); flexibility (F = 25.43; *p* < 0.001; η^2^p = 0.35—large effect), with a significant increase after 8 weeks (*p* = 0.00) and 16 weeks (*p* < 0.001); MILTS (F = 4.90; *p* = 0.01; η^2^p = 0.09—moderate effect), with a significant increase after 8 weeks (*p* = 0.029); the push-up test (F = 18.15; *p* < 0.001; η^2^p = 0.28—large effect), with a significant increase after 8 weeks (*p* < 0.001); abdominal strength–endurance (F = 19.54; *p* < 0.001; η^2^p = 0.30—large effect), with a significant increase after 8 weeks (*p* < 0.001); and the sit-stand test (F = 17.78; *p* < 0.001; η^2^p = 0.26—large effect), with a significant increase after 8 weeks (*p* = 0.00). None of the variables showed differences between the groups or interaction effects between the group and time measurements (*p* > 0.05).

Table 6 also shows the evolution of the cardiorespiratory fitness parameters assessed in the 6MWT during the intervention period. A time effect was observed for the VO_2_peak (F = 10.94; *p* < 0.001; η^2^p = 0.17—large effect), with a significant increase after 8 weeks (*p* = 0.00); distance covered (F = 16.35; *p* < 0.001; η^2^p = 0.25—large effect), with a significant increase after 8 weeks (*p* < 0.001); pre-test DBP (F = 12.23; *p* < 0.001; η^2^p = 0.20—large effect), with a significant reduction after 16 weeks (*p* < 0.001); and final DBP (F = 16.02; *p* < 0.001; η^2^p = 0.24—large effect), with a significant reduction after 16 weeks (*p* < 0.001). None of the variables showed significant differences between the groups, and the interaction effect between the group and time was insignificant (*p* > 0.05).

Table 6 shows the physical and cardiorespiratory tests pre-test and after 8 and 16 weeks of intervention.

### 3.3. Biochemical Parameters

A time effect was observed for the total cholesterol (F = 12.17; *p* < 0.001; η^2^p = 0.18—large effect), with a significant reduction after 8 weeks (*p* < 0.001); LDL-c (F = 18.62; *p* < 0.001; η^2^p = 0.25—large effect), with a significant reduction after 8 weeks (*p* < 0.001); HbA1c (F = 11.71; *p* < 0.001; η^2^p = 0.19—large effect), with a significant reduction after 8 weeks (*p* = 0.03); urea (F = 3.77; *p* = 0.03; η^2^p = 0.07—moderate effect), with a significant reduction after 8 weeks (*p* = 0.04); gamma-glutamyl transferase (F = 7.28; *p* = 0.001; η^2^p = 0.12—moderate effect), with a significant reduction after 16 weeks (*p* = 0.00); lipase (F = 4.47; *p* = 0.01; η^2^p = 0.08—moderate effect), with a significant reduction after 16 weeks (*p* = 0.00); and magnesium (F = 10.30; *p* < 0.001; η^2^p = 0.19—large effect), with a significant reduction after 16 weeks (*p* < 0.001).

Group effects were observed for creatinine (F = 3.18; *p* = 0.04; η^2^p = 0.11—moderate effect), which were significantly higher in the severe/critical group when compared to the moderate group (*p* = 0.04), and CRP (F = 3.46; *p* = 0.01; η^2^p = 0.19—large effect), which were significantly higher in the severe/critical group when compared to the mild group (*p* = 0.01). None of the variables showed an interaction effect between the group and time measurements (*p* > 0.05).

Table 7 shows the responses of biochemical parameters pre-test and after 8 and 16 weeks of intervention.

## 4. Discussion

This study aimed to analyze the effects of multi-professional interventions on health-related physical fitness and biochemical parameters in overweight and obese COVID-19 survivors after 16 weeks of discharge from COVID-19 at different degrees of impairment. The following outcomes were observed: (i) 8 weeks of intervention showed improvements in MIHS values for the right and left sides, MILTS, push-ups, abdominal strength–endurance repetitions, sit and stand test, VO_2_peak, and distance covered in the 6MWT; (ii) increased flexibility values after 8 and 16 weeks; (iii) the pre-test and final DBP showed a significant reduction after 16 weeks and an abdominal circumference after 8 weeks; (iv) the severe group showed an increase in fat mass values and body fat percentage compared to the mild group; (v) 8 weeks showed improvements in the total cholesterol, LDL-c, HbA1c, urea, and in GGT, lipase, and magnesium reduction values after 16 weeks; (vi) the severe group showed an increase in creatinine values compared to the moderate group; (vii) CRP had a significant increase in the severe/critical group compared to the mild group; and (viii) it was observed that the 8- and 16-week interventions showed significant improvements in the body composition parameters, physical fitness, and biomarkers.

In contrast, no significant differences were detected in body mass, BMI, final HR, pre-test SBP, final SBP, HDL-c, TGL, ALT, AST, ALP, albumin, amylase, calcium, and phosphorus. These differences were also not found when investigating intergroup differences (time effect) and the degree of COVID-19 impairment (group effect). There was no significant difference in the BMI between the different groups, although there was a significant difference in the fat mass and body fat percentage, with lower values for the mild group. In addition, no significant differences were observed between the self-reported symptoms of COVID-19 in different degrees; these responses were observed in previous studies that analyzed possible differences under mild, moderate, and severe/critical symptoms in COVID-19 survivors [5,6]. These factors reinforce that some sequels are independent of the disease severity [5,6].

To analyze the outcome of COVID-19, all anthropometric and body composition parameters must be analyzed, not just the BMI alone [6,7]. It is known that excess body fat promotes the secretion of pro-inflammatory mediators, with a consequent reduction in the immune response [10]. Therefore, it is possible to state that being overweight and obese significantly worsens the symptoms of COVID-19 [5,6,7]. A previous study reported that patients with a severe/critical form of COVID-19 showed higher values of fat mass and a body fat percentage than those with a mild form of COVID-19 with the same BMI [7]. Fat mass was significantly higher in severe/critical COVID-19 compared to the mild group; similar data were found by Perli et al. [6] when comparing the different symptoms of the disease. The authors also observed that this difference persisted 1 year after the disease. In this line, regular physical exercise can help control these parameters and promote a better immunological response against COVID-19 infection [38], regardless of the symptoms of the disease. This study did not collect food records before or after the multi-professional interventions. Therefore, it is impossible to establish a relationship between the participant’s body composition and food intake.

People who develop the severe form of COVID-19 need long periods of hospitalization, associated with the frequent use of corticosteroids, the use of prolonged mechanical ventilation, and the use of neuromuscular blockers, causing direct impacts on the musculoskeletal system after hospitalization [39]. The practice of physical activity is considered one of the main components of a healthy life, promoting the prevention of overweight, systemic inflammation, and transmissible viral diseases, proving to be an effective therapeutic strategy to reduce a series of metabolic disorders, thus reducing the effects against the “cytokine storm” reported in patients with COVID-19 [38,39,40]. Regarding health-related physical fitness tests, significant improvements in hand pressure strength were observed for the mild, moderate, and severe/critical COVID-19 groups after 8 weeks of intervention, corroborating the findings of Everaerts et al. [41], who found an improvement in MISH after 16 weeks of post-hospital discharge intervention. The 30 s sit-up and chair–stand test for the lower limbs also showed improvement after 8 weeks of intervention for all COVID-19 groups, reinforcing the findings of Li et al. [42] after 8 and 9 weeks of tele-exercises. MILTS showed an improvement in the time effect after 8 weeks (pre-test vs. post-8w) in the mild, moderate, and severe/critical COVID-19 groups; the present study partially corroborates the results of Sordi et al. [5], showing improvement only in the moderate COVID-19 group.

Cardiorespiratory health was assessed using VO_2_peak, HR, SpO_2_, and blood pressure to verify the physical capacity, effort tolerance, and possible cardiopulmonary changes [5,27]. The VO_2_peak of the mild, moderate, and severe/critical COVID-19 groups after 8 weeks was higher when compared to after 16 weeks of intervention, in line with Rinaldo et al. [43], showing improvement in the VO_2_peak in response to concurrent training after hospital discharge for COVID-19. No significant improvement was detected in SpO_2_. This finding does not corroborate those of Lemos et al. [7], who found an improvement in SpO_2_ values due to the impact of this disease.

Some biochemical analyses did not show significant changes after the intervention: HDL-c, TGL, ALT, AST, ALP, albumin, amylase, calcium, and phosphorus. However, the patient’s biochemical analyses were among normative values at pre-intervention [42,43,44]. Significant changes were verified after intervention for total cholesterol, LDL-c, HbA1c, urea, creatinine, GGT, lipase, magnesium, and CRP. It was observed that dyslipidemia is a risk factor for severe manifestations of COVID-19 [45]. It is known that the lipid profile can change viral infections due to the neutralizing role of lipoproteins, protecting the host [42]; however, in patients with COVID-19, this protection does not appear to occur. After the interventions, there was a significant reduction in the total cholesterol and LDL-c levels after 8 weeks in all groups (mild, moderate, and severe/critical) and an increase in HDL-c levels in the mild and moderate groups compared to the 16 weeks of intervention. This study corroborates other studies that show that the decrease in HDL-c levels correlates with the severity of COVID-19 cases [45,46,47].

Another risk factor for COVID-19 is the increase in serum HbA1c levels. Diabetic individuals are at an increased risk for several infections, including more severe cases of COVID-19 [47,48,49]. A high level of glucose in the bloodstream facilitates the hyper inflammation observed in the cytokine storm [50,51], and the SARS-CoV-2 virus can also cause damage to the pancreatic islets, which are responsible for glucose regulation [50,51]. In the present study, HbA1c levels in the bloodstream were higher in persons with severe COVID-19 even after 16 weeks of intervention; therefore, people with increased blood glucose levels are more likely to progress to severe cases.

Due to the presence of the angiotensin-converting enzyme 2 (ACE2) receptor in several organs, liver dysfunction resulting from COVID-19 may be related to severe infection [49,50,51,52]. Chen et al. [53] observed that ALT, AST, total bilirubin, ALP, and GGT concentrations were higher in deceased persons than in those recovered from COVID-19. Hepatocyte steatosis is derived from the accumulation of lipids in hepatocytes, altering serum levels of triglycerides and HDL-c, and is considered one of the most common causes of chronic liver disease in adults [54]; therefore, interventions with physical exercise are necessary for a better prognosis of HS. GGT levels in this study were reduced only after 16 weeks of intervention in the mild, moderate, and severe/critical COVID-19 groups, showing that just eight weeks is insufficient to improve liver parameters.

Studies on COVID-19 indicate that electrolyte abnormalities, including sodium, potassium, chloride, and calcium abnormalities, are also associated with disease severity in persons with COVID-19 [54,55,56], presenting an association where patients with more severe COVID-19 tend to present hypocalcemia compared to those with less severe forms of the disease [54]. Due to the severity of the disease, many people tend to remain hospitalized for longer, resulting in a loss of muscle mass and bone mineral mass; in females, this process accelerates after menopause due to low estrogen production [56]. In this study, magnesium levels after interventions significantly improved in patients with mild, moderate, and severe/critical COVID-19 after 16 weeks.

There was no statistical difference in serum CRP levels, although there was a significant level reduction when comparing the three intervention moments. At the pre-intervention moment, a high concentration of CRP was found in the experimental groups about the reference values (>5.1 mg/dL), corroborating a previous study that revealed a high concentration of CRP in persons with severe COVID-19 due to the innate system deregulated by the presence of inflammatory cytokines [57,58]. The CRP concentration was reduced in the severe/critical group (pre-test vs. severe/critical) in response to physical exercise, approaching the reference values (<5.0 mg/dL). High levels in the bloodstream can be found in response to active infections or acute inflammatory processes [58,59], but high levels of this marker have been associated with obesity, as, in these persons, the inflammatory response can be precise [58]. Considering the aspects listed, multi-professional interventions that aim to recover the health conditions of overweight and obese people are relevant for promoting health in this significant portion of the Brazilian population, which already has a prevalence of overweight of 61.4% and obesity of 24.3% in people aged 18 or over. Public policies must guide change by integrating multi-professional teams to promote a healthy lifestyle for the better rehabilitation of COVID-19 survivors [5,60].

The absence of significant differences after 16 weeks in the general variables investigated in this research could be related to a lower volume and frequency of physical exercise, i.e., twice a week. The primary physical training adaptations during the first weeks (8 and 12) are related to neural adaptations with subsequent plateaus [61]. Thus, the improvement in the MIHS, MILTS, push-ups, abdominal strength–endurance, and sit-and-reach tests could be explained by neural adaptations after 8 weeks of physical exercise [61]. Considering muscle hypertrophy, the lack of significant differences in skeletal muscle mass and fat-free mass is probably related to a lower volume and lower frequency of strength training [62], in which the main muscle groups must be trained at least twice a week to maximize muscle growth, within a volume and intensity suitable for each person. Detrained people expend more energy doing physical exercises than trained people [63]. Therefore, when detrained people start physical exercises, energy expenditure could be higher in the first weeks, but with time, the expenditure tends to reduce [63]. Given this, the stabilization of the fat mass and body fat percentage after 16 weeks may be justified by a body adaptation [62] or a lower level of manipulation in the volume, frequency, and intensity of the concurrent training [61]. However, considering the health status of the study participants, we were cautious about manipulating some aspects linked to physical exercise.

After 16 weeks, a relevant aspect occurred with the three experimental groups. SBP and DBP were significantly reduced at rest. Pescatello et al. [64] pointed out that chronic exercise may reduce the BP around ~5–7 mmHg with the following mechanisms: a decrease in the cardiac output and/or total peripheral resistance, less sympathetic neural influence, greater local vasodilator influence, higher lumen diameter, and bigger distensibility of the vasculature are structural adaptations to physical training promoting lower peripheral resistance, as well as genetic factors.

Eight limitations can be highlighted in this study: (i) a loss of follow-up between groups because participants did not return for the final assessments; (ii) the application of the food record before the intervention, after 8 and 16 weeks; (iii) the absence of other biochemical measures, such as pro-inflammatory cytokines and anti-inflammatory cytokines, coagulation factors, and cardiac markers; (iv) the monitoring of the body composition and biomarkers after 16 weeks of this study to verify whether changes in eating habits and physical activity were persistent; (v) to perform fine-tuning in the volume, intensity, and frequency of physical exercise for study participants; (vi) including normal-weight participants to investigate these same relationships in a “healthy” comparison group; (vii) difficulty in composing a control group, as many persons were asymptomatic (another problem in recruiting a control group at this stage of COVID-19 in Brazil refers to providing care to the control group since these people could contract SARS-CoV-2 at any time, and another point would be not to assist survivors of COVID -19, as this would be unethical and could promote worse effects on the biopsychosocial health of these survivors); and (viii) include normal-weight participants to investigate these same relationships in a “healthy” comparison group.

Another point that needs to be considered is dropouts (health problems; *n =* 9) from the present study related to low %SpO_2_ during physical exercise and at rest and medical advice for suspending practical activities. This study highlights the importance of a 16-week multi-professional intervention for addressing COVID-19 sequelae in overweight or obese individuals, following WHO guidelines [16]. It emphasizes the role of concurrent training combined with a team of physicians, nutritionists, exercise physiologists, psychologists, physiotherapists, and biomedical professionals in aiding the recovery and resumption of daily activities. The study suggests that future research should focus on long-term monitoring, differences in response by sex and age, and exploring various training combinations (aerobic vs. resistance) to understand pathophysiological responses better. Additionally, including a control group without the disease could enhance the effectiveness of rehabilitation strategies.

The severity of COVID-19 symptoms directly influences the progression of physical exercise programs, necessitating individualized training based on initial evaluations. Professionals must conduct comprehensive assessments, including physical fitness and recent blood results, to devise effective strategies for improving quality of life. Addressing individual needs and sequelae is crucial for recovering health-related physical fitness in COVID-19 survivors. Continuous monitoring and follow-up are essential and urgent. This study underscores the importance of developing strategies to improve health conditions through physical activity, nutrition, and psychoeducation, aiming for comprehensive care and support for COVID-19 survivors.

## 5. Conclusions

This study concludes that an 8-week intervention significantly improved the anthropometrics, body composition, and physical fitness and influenced reductions in the lipid profile and glycated hemoglobin. After 16 weeks, these variables stabilized, indicating lasting benefits. The multi-professional intervention model proved beneficial for post-COVID-19 patients, regardless of the symptom severity. The study’s clinical trial design, incorporating patients with varying COVID-19 symptoms, enhances the validity and broadens the understanding of a multi-professional program’s effects. This approach combines nutritional education to promote healthy eating, psychoeducation for knowledge and behavior changes, and biochemical tests to assess the impacts on overweight and obese individuals.

## 6. Practical Applications

To thoroughly assess various health indicators—such as health-related physical fitness, vital signs, nutritional aspects, biochemical markers, and mental health—and to conduct a detailed medical history is essential for understanding the potential persistent sequelae of COVID-19. This comprehensive evaluation is crucial for guiding health recovery efforts for these patients. Given the heterogeneous nature of complaints and symptoms, a skilled team’s individualized direction of multi-professional interventions is indispensable for recovering post-COVID-19 patients. Prolonged interventions exceeding 16 weeks are recommended to improve various biopsychosocial aspects of these patients. Furthermore, periodic evaluations can more accurately guide rehabilitation programs for post-COVID patients. A significant portion of patients with post-COVID syndrome continue to experience physical and psychological limitations that hinder their ability to perform daily and occupational activities. Therefore, promoting multidisciplinary interventions is imperative and urgent for a substantial population affected by COVID-19.

## Figures and Tables

**Figure 1 healthcare-12-02034-f001:**
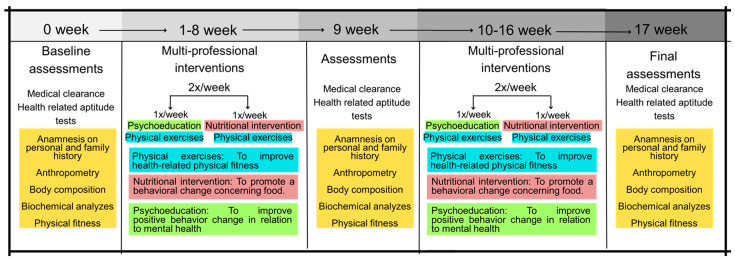
Methodological design of the present study.

**Figure 2 healthcare-12-02034-f002:**
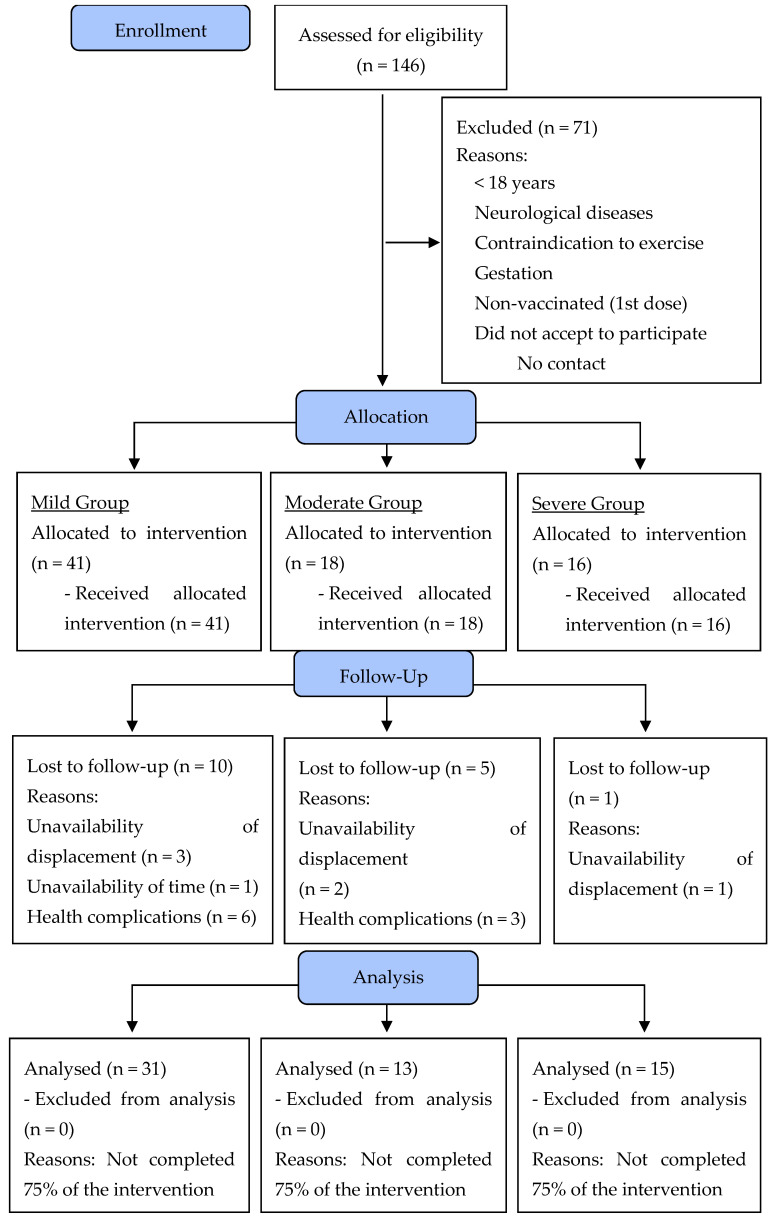
Flowchart diagram of the participants of the present study.

**Table 1 healthcare-12-02034-t001:** Nutritional activities that were developed throughout the research project.

Order	Theme	Details
I	Pre- and post-exercise	The theme explores pre- and post-exercise nutrition, emphasizing the importance of diet in exercise performance. It includes examples of beneficial foods, recommended quantities, and the optimal response time after consumption.
II	Introduction to healthy eating	Food builders, regulators, energy foods, and food pyramid; healthy eating explains the builders, regulators, and energy of the different foods, their due quantities, and their position in the food pyramid.
III	How to assemble a healthy dish	The lesson details the appropriate quantities of each food group—carbohydrates, proteins, fats, salads, and vegetables—and their portions for each meal. It includes a practical demonstration of how to assemble these on a plate.
IV	Gain of muscle mass	The theme focuses on incorporating foods into meals that support muscle mass maintenance and growth. The class provides examples of protein sources and emphasizes the importance of gaining and maintaining muscle mass.
V	Micronutrients (vitamins and minerals)	The class of micronutrients: the importance of vitamins and minerals in adolescent health, nutritional interactions, and examples of where we come from in food.
VI	Soluble and insoluble fiber	The class on fibers covers the importance of daily fiber intake, the recommended amounts, the differences between soluble and insoluble fibers, and the food sources for each type.
VII	How to read food labels	Food labels: how to read food labels, plus practical examples such as sachet juices, biscuits, and processed foods.
VIII	Types of hunger (emotional, regulatory, specific, and social)	Physical or emotional hunger: explain in detail how to identify the hunger level and whether it is physical or emotional.
IX	Intermediate snacks	This theme explains their importance, the necessary amount, and examples that can be applied to daily routines.
X	Stress and anxiety	The lesson covers stress and anxiety, explaining their definitions and how to identify associated symptoms and physical and psychological signs. The intervention emphasizes self-awareness and managing these conditions through practical breathing exercises.
XI	How to deal with post-COVID-19 sequelae	Due to the various sequelae after COVID-19, the class intends to work on the possible sequelae after the virus and how nutrition can help treat and decrease symptoms.
XII	Mindful eating	The class cultivates participants’ ability to practice mindful eating and develop self-control in managing thoughts, emotions, physical sensations, and eating habits.
XIII	Diet, light, and typical foods	The class explains the different types of foods and the time each is eaten and gives examples of each product.
XIV	Myths and truths of nutrition	To address common myths about weight loss, such as “water fasting with lemon aids slimming”, “sweating leads to weight loss”, and “cutting carbohydrates helps lose weight”, among others.
XV	Resume the topics already covered	Through a conversation circle, all topics were discussed. The aim was to create an environment to clarify doubts and revisit previously covered subjects.
XVI	Ten steps to healthy eating	The last lesson addressed the ten steps to healthy eating and how to maintain these habits during vacation, independent of the research group.

**Table 2 healthcare-12-02034-t002:** Psychological activities that were developed throughout the research project.

Order	Theme	Details
I	Introduction to mental health	This meeting outlined the rules, participants’ duties, reasons for the mental health interventions, and the topics to be covered.
II	Exercise and mental health	Reporting the importance of physical exercise in motivating participants about the benefits of exercise for a better quality of life and mental health.
III	How to assemble a healthy dish	In this meeting, we addressed anxiety in daily life, its impact on activities, ways to identify it, and techniques for managing it based on cognitive–behavioral theory.
IV	The obesity today	The round table discussion clarified misconceptions, biases, and stereotypes about obesity, aiming to illustrate to participants that it is a multifactorial disease.
V	Food and comportaments	Explain to the participants how food is directly linked to our feelings, explaining the types of hunger: psychological, physical, and social.
VI	Post-traumatic stress disorder	This intervention aimed to help participants identify the disorder’s symptoms; this was completed by presenting and explaining each symptom.
VII	Promoting a healthy lifestyle	Using daily examples, videos, and discussion circles, participants learned to prioritize physical and mental well-being for a balanced life, regardless of their circumstances.
VIII	Reflections on stress	To help participants understand how stress can affect their mental and physical health. In addition, it will show techniques and behaviors to relieve this feeling healthily.
IX	Reflections on depressive symptoms	Demonstrating the types of depression and their common symptoms helps identify them while proposing techniques and behaviors for prevention and treatment.
X	Insomnia and relaxation techniques.	Explain how good quality sleep can influence physical and mental health. In addition, explain how sleep hygiene can help maintain a good rest and recovery routine.
XI	Reflections on denial	The topic aimed to explain how harmful defense mechanisms influence behaviors and choices.
XII	Reflections on fear	To show how fear can affect our feelings and interpersonal relationships and demonstrate examples from everyday life.
XIII	Binge eating	Define, demonstrate, and explain how this disorder works and ways of identifying it.
XIV	Behavior change	A critical sense was developed to perceive inappropriate behavior and change it.
XV	Reflections on bereavement	The stages of bereavement were presented, and there was a space for people to talk about these events.
XVI	Reflections on aging	The main changes regarding aging were shown through a content presentation.

**Table 3 healthcare-12-02034-t003:** Physical exercise program for mild, moderate, and severe/critical COVID-19 survivors.

Order	Training Program A	Training Program B
1	Warm-up (walking, cycling, or rowing at moderate intensity for 8 min)	Warm-up (walking, cycling, or rowing at moderate intensity for 8 min)
2	Plank torso strength	Plank torso strength
3	Rectus abdominais	Rectus abdominais
4	Aerobic exercise	Hip bridge
5	Squat	Leg press
6	Leg extension	Aerobic exercise
7	Bench press	Leg curl
8	Aerobic exercise	Push up
9	Cable pulldown	Cable straight-back seated row
10	Dumbbell shoulder press	Front raise
11	Triceps pulley	Biceps curl
12	Aerobic exercise	Aerobic exercise

**Table 4 healthcare-12-02034-t004:** Clinical characteristics of patients of three COVID-19 survivors intervention groups.

Variables	Mild (*n =* 31)	Moderate (*n =* 13)	Severe/Critical (*n =* 15)	*p*-Value
Age (years old)	53.2 ± 12.3	54.3 ± 15.0	50.9 ± 12.9	*p* = 0.77
Sex				*p* = 0.33
Male	18 (58.1%)	4 (30.8%)	10 (66.7%)	
Female	13 (41.9%)	9 (69.2%)	5 (33.3%)	
BMI (kg/m^2^)	29.5 ± 4.8	31.1 ± 6.2	32.7 ± 4.8	*p* = 0.14
Medical history				
Hypertension	10 (32.3%)	5 (38.5%)	8 (53.3%)	*p* = 0.40
Diabetes	6 (19.4%)	2 (15.4%)	6 (40.0%)	*p* = 0.23
Dyslipidemia	9 (29.0%)	1 (7.7%)	3 (20.0%)	*p* = 0.30
Hypothyroidism	4 (12.9%)	4 (30.8%)	3 (20.0%)	*p* = 0.39
Psychogenic change	9 (29.0%)	2 (15.4%)	0 (0%)	*p* = 0.06
Neuropathy	2 (6.3%)	0 (0%)	2 (13.3%)	*p* = 0.39
Asthma	0 (0%)	1 (7.7%)	0 (0%)	*p* = 0.17
Heart disease	6 (19.4%)	0 (0%)	6 (40.0%)	*p* = 0.03
**Post-COVID-19 self-reported symptoms**				
Fatigue	13 (41.9%)	7 (53.8%)	7 (46.7%)	*p* = 0.78
Dyspnoea	2 (6.3%)	3 (23.1%)	1 (6.7%)	*p* = 0.23
Muscle pain	10 (32.3%)	6 (46.2%)	8 (53.3%)	*p* = 0.37
Joint pain	0 (0%)	0 (0%)	1 (6.7%)	*p* = 0.23
Cough	7 (22.6%)	2 (15.4%)	4 (26.7%)	*p* = 0.78
Dizziness	6 (19.4%)	4 (30.8%)	5 (33.3%)	*p* = 0.54
Memory deficit	22 (71.0%)	9 (69.2%)	9 (60.0%)	*p* = 0.76
Difficulty concentrating	12 (38.7%)	4 (30.8%)	5 (33.3%)	*p* = 0.87
Anxiety disorder	10 (32.3%)	6 (46.2%)	7 (46.7%)	*p* = 0.55
Depression	4 (12.9%)	4 (30.8%)	2 (13.3%)	*p* = 0.33
Processing speed	12 (38.7%)	6 (46.2%)	4 (26.7%)	*p* = 0.57
Physical activity ≥150 min/week	9 (29.0%)	2 (15.4%)	6 (40.0%)	*p* = 0.37
Baseline vital signs				
HR (bpm)	78 ± 10	85 ± 15	81 ± 10	*p* = 0.18
SBP (mmHg)	127 ± 12	127 ± 10	128 ± 17	*p* = 0.95
DBP (mmHg)	78 ± 16	82 ± 9	81 ± 17	*p* = 0.61
% SpO₂	97.0 ± 1.6	97.4 ± 1.4	96.8 ± 1.6	*p* = 0.62

Note: numerical data are expressed as mean ± standard deviation, and categorical data are expressed as absolute and relative frequency (%); BMI = body mass index; HR = heart rate; SBP = systolic blood pressure; DBP = diastolic blood pressure; %SpO_2_ = oxygen saturation; significance level *p* < 0.05.

**Table 5 healthcare-12-02034-t005:** Anthropometry and body composition response pre-test after 8 and 16 weeks of intervention in the three COVID-19 survivors’ groups.

Variables	Mild (*n =* 31)			Moderate (*n =* 13)			Severe/Critical (*n =* 15)		
	Pre-Test	Post-8W	Post-16W	Pre-Test	Post-8W	Post-16W	Pre-Test	Post-8W	Post-16W
Body mass (kg)	84.5 ± 19.4 (77–92)	84.5 ± 19.0 (77–92)	82.7 ± 22.2 (75–90)	80.5 ± 16.1 (69–91)	80.6 ± 15.8 (69–92)	80.7 ± 15.9 (69–92)	100.1 ± 23.1 (90–110)	95.3 ± 27.7 (84–106)	99.1 ± 22.1 (88–110)
BMI (kg/m^2^)	29.6 ± 4.9 (28–31)	29.6 ±4.8 (28–31)	29.5 ± 4.7 (28–31)	31.0 ± 5.9 (28–34)	31.1 ± 6.0 (28–34)	31.1 ± 6.2 (28–34)	33.2 ± 5.3 (30–36)	32.9 ± 5.0 (30–35)	32.7 ± 4.8 (30–35)
AC (cm) *	101.3 ± 13.6 (96–106)	100.5 ± 13.4 (96–105)	100.5 ± 13.2 (96–105)	101.8 ± 14.3 (94.0–110)	100.7 ± 13.8 (93–108)	100.4 ± 14.2 (93–108)	111.0 ± 14.9 (104–118)	108.7 ± 12.1 (102–115)	109.2 ± 14.0 (102–116)
FFM (kg)	53.0 ± 12.2 (49–57)	53.4 ± 12.2 (49–58)	53.9 ± 13.0 (49–58)	46.2 ± 10.4 (40–53)	45.9 ± 9.9 (39–53)	46.1 ± 10.0 (39–53)	57.6 ± 13.0 (51–64)	55.7 ± 13.9 (49–62)	59.1 ± 13.8 (53–66)
SMM (kg)	29.4 ± 7.2 (27–32)	29.6 ± 7.2 (27–32)	29.9 ± 7.7 (27–33)	25.4 ± 6.3 (21–29)	25.2 ± 6.0 (21–29)	25.3 ± 6.1 (21–29)	32.1 ± 7.7 (28–36)	32.2 ± 7.7 (28–36)	33.0 ± 8.1 (29–37)
FM (kg) ‡	31.5 ± 11.5 (27–36)	31.2 ± 10.8 (27–35)	32.0 ± 11.6 (28–36)	34.3 ± 10.7 (28–41)	34.8 ± 11.0 (28–41)	34.6 ± 10.7 (28–41)	42.6 ± 13.3 (36–49)	41.6 ± 13.0 (36–47)	40.0 ± 11.3 (34–46)
BFP (%) ‡	36.8 ± 8.1 (34–39)	36.5 ± 7.8 (34–39)	36.1 ± 8.5 (33–39)	42.1 ± 8.4 (38–46)	42.5 ± 8.7 (38–47)	42.3 ± 8.3 (38–47)	43.5 ± 5.2 (39–48)	41.4 ± 7.3 (37–45)	40.1 ± 6.6 (36–44)

Note: Data described by the mean, standard deviation (±), and 95% confidence intervals (CI); W = weeks; BMI = body mass index; AC = abdominal circumference; FFM = fat-free mass; SMM = musculoskeletal mass; FM = fat mass; BFP = body fat percentage; * = time effect (*p* < 0.05, pre-test vs. post-8W); ‡ = group effect (*p* < 0.05, mild vs. severe/critical).

**Table 6 healthcare-12-02034-t006:** Physical and cardiorespiratory fitness responses pre-test and after 8 and 16 weeks of intervention in the three COVID-19 survivors’ groups.

Variables	Mild (*n =* 31)			Moderate (*n =* 13)			Severe/Critical (*n =* 15)		
	Pre-Test	Post-8W	Post-16W	Pre-Test	Post-8W	Post-16W	Pre-Test	Post-8W	Post-16W
MIHS-R (kgf) *	30 ± 11 (26–35)	35 ± 12 (30–39)	35 ± 11 (30–39)	29 ± 13 (22–35)	30 ± 12 (23–36)	31 ± 12 (25–38)	33 ± 13 (26–39)	36 ± 11 (29–42)	37 ± 13 (30–43)
MIHS-L (kgf) *	28 ± 11 (25–32)	32 ± 11 (28–37)	34 ± 11 (29–38)	25 ± 8 (19–31)	29 ± 12 (22–35)	30 ± 12 (23–37)	33 ± 13 (27–38)	34 ± 12 (29–38)	35 ± 13 (29–41)
Sit and reach (cm) §	22 ± 9 (19–25)	24 ± 9 (21–27)	27 ± 8 (24–30)	27 ± 6 (22–32)	30 ± 6 (25–35)	31 ± 4 (26–35)	19 ± 9 (15–24)	21 ± 10 (17–26)	24 ± 9 (20–28)
MILTS (kg) *	88 ± 38 (75–102)	100 ± 23 (86–114)	104 ± 35 (91–118)	69 ± 20 (47–91)	85 ± 23 (64–107)	84 ± 28 (63–104)	106 ± 49 (84–127)	108 ± 42 (89–128)	108 ± 45 (89–127)
Push-up (reps/min) *	19 ± 9 (16–22)	24 ± 11 (21–28)	28 ± 14 (24–33)	15 ± 7 (10–20)	23 ± 8 (18–28)	25 ± 10 (19–32)	15 ± 8 (10–20)	16 ± 7.0 (11–21)	18 ± 8 (11–24)
Abdominal strength–endurance (reps) *	17 ± 8 (14–20)	21 ± 8 (17–25)	26 ± 12 (22–30)	16 ± 8 (11–21)	19 ± 8 (14–24)	24 ± 11 (17–30)	14 ± 9 (10–19)	20 ± 11 (15–25)	19 ± 10 (13–24)
Sit and stand (reps/min) *	16.3 ± 4.4 (14–18)	19.1 ± 5.9 (17–21)	21.6 ± 5.6 (20–24)	15.8 ± 4.2 (13–19)	20.0 ± 5.9 (17–23)	19.9 ± 4.5 (17–23)	15.8 ± 6.3 (13–18)	17.7 ± 5.6 (15–21)	18.2 ± 5.2 (15–21)
6MWT									
VO_2_ peak (mL/kg/min) *	16.9 ± 3.6 (15–18)	18.0 ± 3.8 (16–20)	18.6 ± 4,0 (17–20)	16.3 ± 5.4 (14–19)	17.3 ± 5.0 (15–20)	17.7 ± 4.8 (15–20)	16.6 ± 3.8 (14–19)	17.4 ± 4.5 (15–20)	17.8 ± 3.5 (15–20)
Distance (m) *	534 ± 69 (502–566)	583 ± 89 (546–620)	595 ± 96 (561–630)	505 ± 110 (455–555)	561 ± 99 (507–615)	562 ± 90 (512–613)	521 ± 108 (473–570)	569 ± 109 (519–619)	554 ± 80 (505–602)
Final heart rate (bpm)	78 ± 10 (74–82)	80 ± 10 (76–84)	76 ± 11 (70–81)	80 ± 10 (75–86)	72 ± 9 (67–78)	77 ± 11 (71–83)	79 ± 10 (74–84)	76 ± 11 (70–81)	76 ± 9 (71–82)
SBP pre-test (mmHg)	125 ± 13 (120–129)	124 ± 8 (118–129)	123 ± 12 (118–128)	128 ± 13 (121–135)	132 ± 19 (124–140)	131 ± 14 (124–139)	126 ± 12 (119–132)	134 ± 15 (127–141)	128 ± 15 (123–135)
SBP final (mmHg)	142 ± 16 (135–148)	148 ± 11 (142–153)	141 ± 18 (135–148)	141 ± 21 (130–151)	148 ± 17 (141–156)	143 ± 18 (133–152)	148 ± 23 (138–158)	147 ± 11 (141–154)	145 ± 12 (136–154)
DBP pre-test (mmHg) ‡	79 ± 9 (75–83)	81 ± 11 (77–85)	71 ± 9 (68–75)	79 ± 13 (73–85)	83 ± 8 (77–89)	75 ± 11 (70–81)	79 ± 12 (73–85)	87 ± 12 (81–92)	77 ± 10 (72–82)
DBP final (mmHg) ‡	84 ± 11 (80–89)	87 ± 13 (83–92)	75.5 ± 10.9 (71–80)	86 ± 14 (79–93)	91 ± 9 (84–97)	76 ± 11 (70–83)	86 ± 15 (79–92)	87 ± 12 (80–93)	79 ± 13 (73–85)

Note: Data described by mean, standard deviation (±), and 95% confidence intervals (CI); W = weeks; SBP = systolic blood pressure; DBP = diastolic blood pressure; MIHS = maximal isometric handgrip strength; R = right side; L = left side; MILTS = maximal isometric lumbar-traction strength; 6MWT = 6 min walk test; VO_2_peak = peak oxygen consumption. * = time effect (*p* < 0.05, pre-test vs. post-8W); ‡ = time effect (*p* < 0.05, pre-test vs. post-16W), and § = time effect (*p* < 0.05; pre-test vs. post-8W and post-16W).

**Table 7 healthcare-12-02034-t007:** Biochemical parameter responses pre-test and after 8 and 16 weeks of intervention in the three COVID-19 survivors’ groups.

Variables	Mild (*n =* 31)			Moderate (*n =* 13)			Severe/Critical (*n =* 15)		
	Pre-Test	Post-8W	Post-16W	Pre-Test	Post-8W	Post-16W	Pre-Test	Post-8W	Post-16W
TC (mg/dL) *	182 ± 48 (163–201)	165 ± 35 (151–178)	168 ± 35 (155–182)	198 ± 56 (168–227)	152 ± 43 (131–172)	159 ± 32 (139–180)	189 ± 61 (161–216)	158 ± 35 (140–177)	157 ± 42 (138–176)
LDL-c (mg/dL) *	119 ± 52 (99–139)	88 ± 28 (77–98)	88 ± 29 (76–100)	125 ± 53 (94–156)	74 ± 34 (57–91)	82 ± 27 (63–101)	115 ± 64 (86–144)	85 ± 30 (69–100)	83 ± 45 (65–100)
HDL-c (mg/dL)	49.6 ± 12.9 (45.2–54.0)	52.4 ± 15.1 (48.0–56.8)	56.0 ± 11.2 (51.6–60.3)	54.2 ± 14.3 (47.4–60.9)	50.0 ± 15.1 (43.2–56.9)	53.9 ± 15.4 (47.2–54.8)	48.1 ± 7.3 (41.6–54.6)	44.6 ± 10.0 (38.2–51.0)	48.5 ± 10.9 (42.2–54.8)
TGL (mg/dL)	118 ± 50 (98–138)	124 ± 54 (104–144)	115 ± 54 (92–137)	130 ± 59 (99–161)	112 ± 46 (80–144)	106 ± 45 (70–142)	134 ± 65 (105–163)	142 ± 63 (113–171)	142 ± 85 (110–174)
HbA1c (%) *	6.1 ± 0.6 (5.9–6.4)	6.0 ± 0.7 (5.6–6.5)	5.6 ± 0.4 (5.4–5.8)	6.3 ± 1.2 (5.9–6.7)	6.0 ± 1.0 (5.6–6.5)	5.7 ± 0.5 (5.4–5.9)	5.8 ± 0.5 (5.4–6.2)	5.7 ± 0.6 (5.2–6.1)	5.6 ± 0.4 (5.4–5.9)
Creatinine (mg/dL) †	1.2 ± 0.2 (1.2–1.3)	1.2 ± 0.3 (1.1–1.3)	1.2 ± 0.3 (1.1–1.3)	1.1 ± 0.2 (1.0–1.3)	1.0 ± 0.4 (0.9–1.2)	1.2 ± 0.2 (1.0–1.3)	1.3 ± 0.2 (1.1–1.4)	1.3 ± 0.2 (1.1–1.4)	1.4 ± 0.2 (1.2–1.5)
Urea (mg/dL) *	38.4 ± 15.7 (32.9–43.8)	35.2 ± 12.9 (30.8–39.6)	41.2 ± 11.4 (37.3–45.1)	40.9 ± 17.3 (32.4–49.3)	29.8 ± 13.5 (23.1–36.4)	37.0 ± 11.5 (30.9–43.1)	38.1 ± 11.4 (30.2–45.9)	32.1 ± 7.5 (25.7–38.6)	31.7 ± 9.2 (25.9–37.6)
ALT (U/L)	27.5 ± 10.2 (23.7–31.2)	27.7 ± 13.2 (22.6–32.8)	29.0 ± 13.4 (24.9–33.1)	25.7 ± 12.2 (20.0–31.4)	26.2 ± 13.2 (18.4–34.1)	24.2 ± 8.6 (17.9–30.6)	29.4 ± 8.8 (24.1–34.8)	33.9 ± 16.8 (26.6–41.3)	22.5 ± 8.5 (16.6–28.4)
AST (U/L)	29.5 ± 10.6 (25.2–33.9)	29.2 ± 10.5 (25.1–33.3)	27.1 ± 9.4 (23.5–30.7)	22.5 ± 9.0 (16.0–29.1)	34.1 ± 16.6 (27.7–40.4)	25.8 ± 10.1 (20.2–31.4)	31.1 ± 15.4 (25.1–37.2)	27.9 ± 7.3 (22.0–33.9)	24.6 ± 11.3 (19.4–29.8)
ALP (U/L)	51.3 ± 19.5 (43.9–58.8)	59.9 ± 21.3 (52.7–67.1)	57.0 ± 11.8 (51.4–62.6)	59.8 ± 22.8 (48.3–71.3)	54.5 ± 20.5 (43.4–65.6)	68.3 ± 22.7 (59.9–76.7)	60.1 ± 21.3 (49.4–70.8)	59.7 ± 16.2 (52.7–67.1)	54.5 ± 12.6 (46.7–62.3)
GGT (U/L) §	45.7 ± 21.3 (39–52)	39.5 ± 19.4 (33–46)	35.3 ± 16.1 (30–41)	43.7 ± 13.7 (33–55)	43.0 ± 19.8 (32–54)	36.8 ± 14.9 (28–45)	48.0 ± 17.3 (38–58)	43.6 ± 18.2 (34–53)	31.7 ± 11.6 (24–39)
Albumin (g/dL)	4.2 ± 0.6 (3.9–4.4)	4.1 ± 0.4 (3.9–4.2)	4.0 ± 0.6 (3.8–4.2)	3.9 ± 0.7 (3.5–4.2)	4.1 ± 0.4 (3.9–4.3)	4.1 ± 0.7 (3.7–4.4)	4.1 ± 0.4 (3.8–4.4)	4.1 ± 0.3 (3.9–4.3)	4.2 ± 0.8 (3.8–4.5)
Amylase (U/L)	82.8 ± 34.2 (71.7–94.0)	76.7 ± 24.1 (67.2–86.3)	65.3 ± 29.3 (54.5–76.1)	65.5 ± 29.0 (48.3–82.8)	59.1 ± 32.8 (44.3–73.9)	55.8 ± 17.8 (391–72.4)	62.7 ± 24.7 (46.7–78.8)	63.3 ± 26.1 (49.6–77.1)	75.9 ± 38.6 (60.4–91.4)
Lipase (U/L) ‡	53.0 ± 18.3 (47–59)	47.6 ± 12.2 (43–52)	38.3 ± 11.8 (33–43)	45.2 ± 16.8 (36–54)	40.3 ± 14.3 (33–47)	39.7 ± 14.2 (32–47)	45.1 ± 11.6 (36–54)	50.9 ± 10.2 (45–57)	45.3 ± 15.7 (38–52)
Calcium (mg/dL)	6.4 ± 3.6 (5.3–7.6)	6.2 ± 2.5 (5.2–7.2)	6.9 ± 3.1 (5.7–8.0)	6.0 ± 2.5 (4.2–7.8)	7.0 ± 2.8 (5.4–8.6)	8.8 ± 3.6 (7.1–10.6)	6.8 ± 3.0 (5.2–8.5)	7.1 ± 3.5 (5.7–8.6)	5.8 ± 3.0 (4.2–7.4)
Magnesium (mg/dL) §	2.2 ± 0.6 (1.9–2.5)	2.1 ± 0.8 (1.8–2.4)	1.6 ± 0.6 (1.3–1.8)	2.1 ± 1.2 (1.7–2.6)	1.9 ± 0.7 (1.5–2.4)	1.9 ± 1.1 (1.5–2.4)	2.3 ± 0.6 (1.9–2.7)	2.3 ± 0.9 (1.9–2.7)	1.4 ± 0.6 (0.9–1.8)
Phosphorus (mg/dL)	2.5 ± 0.8 (2.1–2.9)	3.1 ± 1.0 (2.8–3.4)	3.5 ± 1.0 (3.2–3.8)	2.9 ± 1.1 (2.3–3.5)	2.9 ± 0.6 (2.4–3.4)	3.6 ± 0.2 (3.0–4.1)	3.2 ± 1.5 (2.6–3.8)	3.0 ± 0.8 (2.5–3.4)	2.9 ± 0.6 (2.5–3.4)
CRP (mg/dL) ‡	6.1 ± 6.9 (3.3–8.9)	6.2 ± 5.0 (4.1–8.3)	5.2 ± 4.6 (3.3–7.0)	9.6 ± 8.6 (5.3–14.0)	8.9 ± 7.2 (5.3–12.4)	5.9 ± 4.7 (3.1–8.7)	13.5 ± 7.5 (9.3–17.6)	10.9 ± 5.5 (7.8–14.0)	7.0 ± 6.2 (4.4–9.6)

Note: Data described by mean, standard deviation (±), and 95% confidence intervals (CI); TC = total cholesterol; LDL-c = LDL cholesterol; HDL-c = HDL cholesterol; TGL = triglycerides; HbA1c = glycated hemoglobin; ALT = alanine aminotransferase; AST = aspartate aminotransferase; ALP = alkaline phosphatase; GGT = gamma-glutamyl transferase; CRP = C-reactive protein; * = time effect (*p* < 0.05, pre-test vs. post-8W); § = time effect (*p* < 0.05, pre-test vs. post-16W); † = group effect (*p* < 0.05, moderate vs. severe/critical), and ‡ = group effect (*p* < 0.05, mild vs. severe).

## Data Availability

The data generated during the study will be informed when requested.

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
