# Peer review of "Impact of Multi-Professional Intervention on Health-Related Physical Fitness and Biomarkers in Overweight COVID-19 Survivors for 8 and 16 Weeks: A Non-Randomized Clinical Trial"

_healthcare, 2024, doi:10.3390/healthcare12202034_

Round 1

Reviewer 1 Report (Previous Reviewer 1)

Comments and Suggestions for Authors

Dear all,

I would like to start by thanking you for the opportunity to review this manuscript. The manuscript fits with the aim of Healthcare, and the subject reveals good content for researchers and professionals in the field.,

However, authors have already resolved most of the issues in previous revision, but there are still minor issues below:

Materials and Methods

Doing 10 measures in one day (the 2nd day), 6 of them are physical tests for overweight and obese COVID-19 survivors, I am afraid that will affect negatively the measures outcomes?

Didn’t use a control group should be added to the limitation.

Figures 1&2. Still the resolution is not sufficient; figures need optimizing display size.

Results

I wish if the authors can present the differences between males and females for (anthropometry, body composition, physical and cardiorespiratory fitness, and biochemical parameters after the intervention) this will provide helpful information about gender differences.

Best regards,

Best wishes,

Author Response

Reviewer 1:

Dear all,

I would like to start by thanking you for the opportunity to review this manuscript. The manuscript fits with the aim of Healthcare, and the subject reveals good content for researchers and professionals in the field.

Re: Thank you for your feedback.

However, authors have already resolved most of the issues in previous revision, but there are still minor issues below:

Materials and Methods

Doing 10 measures in one day (the 2nd day), 6 of them are physical tests for overweight and obese COVID-19 survivors, I am afraid that will affect negatively the measures outcomes?

Re: In a previous study of our research group, we tested this protocol with positive responses to the tests performed. In the article, we already inserted this information about this content: “The physical fitness tests were performed following the recommendations published earlier by Sordi et al. [5], in a clinical trial with post-COVID-19 patients.”

Didn’t use a control group should be added to the limitation.

Re: Thank you for your appointment. We have yet inserted this information in the limitation section: Eight limitations can be highlighted in this study: (i) loss of follow-up between groups because participants did not return for final assessments; (ii) application of the food record before the intervention, after 8 and 16 weeks; (iii) absence of other biochemical measures, such as pro-inflammatory cytokines and anti-inflammatory cytokines, coagu-lation factors and cardiac markers; (iv) monitoring of body composition and biomarkers after 16 weeks of this study to verify whether changes in eating habits and physical activ-ity were persistent; (v) to perform fine-tuning in volume, intensity and frequency of physical exercise for study participants; (vi) include a normal-weight participant to investigate these same relationships in a "healthy" comparison group; (vii) difficulty in composing a control group, as many persons are asymptomatic (another problem in recruiting a con-trol group at this stage of COVID-19 in Brazil refers to providing care to the control group since these people could contract SARS-CoV-2 at any time, and another point would be not to assist survivors of COVID -19, as this would be unethical and could promote worse effects on the biopsychosocial health of these survivors); and (viii) include a normal-weight participant to investigate these same relationships in a "healthy" comparison group.

Figures 1&2. Still the resolution is not sufficient; figures need optimizing display size.

Re: Ok, we adjusted the pixels.

Results

I wish if the authors can present the differences between males and females for (anthropometry, body composition, physical and cardiorespiratory fitness, and biochemical parameters after the intervention) this will provide helpful information about gender differences.

Re: Considering that our study presents much information, we inserted this information in tables in the supplementary document. We organized tables with males and females (without symptoms) and tables just for males and females stratified by sex. Given this, nine tables about this content were inserted in the supplementary material.

Reviewer 2 Report (Previous Reviewer 3)

Comments and Suggestions for Authors

Dear Authors,

Thank you for revising the manuscript and for your thoughtful responses. I appreciate your efforts to enhance the quality of the work. Below are my minor comments for further refinement:

Abstract

Line 21: Please emphasize the importance of the study more compellingly to engage readers.

Line 29: Consider removing effect sizes throughout the abstract, as it is already lengthy.

Additionally, please include a recommendation statement to guide future research or practice.

Introduction

I suggest incorporating recent publications to strengthen the context and relevance of your study. This will help highlight current trends and gaps in the literature.

Materials and Methods

The methods section shows noticeable improvement compared to the first version, congratulations on this enhancement.

Results & Discussion

I am satisfied with the presentation of the study results and the discussion section. The authors have effectively addressed my previous comments, resulting in a notable improvement in the overall quality of the manuscript.

Comments on the Quality of English Language

 Minor editing of English language required.

Author Response

Reviewer 2:

Dear Authors,

Thank you for revising the manuscript and for your thoughtful responses. I appreciate your efforts to enhance the quality of the work. Below are my minor comments for further refinement:

Re: Thank you for your feedback.

Abstract

Line 21: Please emphasize the importance of the study more compellingly to engage readers.

We changed it to: “Considering the diverse symptomatology of COVID-19—ranging from mild to severe cases—multi-professional interventions are crucial for enhancing physical recovery, nutritional status, and mental health outcomes in affected patients.”

Line 29: Consider removing effect sizes throughout the abstract, as it is already lengthy.

Re: it was deleted.

Additionally, please include a recommendation statement to guide future research or practice.

Re: We inserted the practical applications topic:

  1. Practical applications

To thoroughly assess various health indicators—such as health-related physical fitness, vital signs, nutritional aspects, biochemical markers, and mental health—and to conduct a detailed medical history is essential for understanding the potential persistent sequelae of COVID-19. This comprehensive evaluation is crucial for guiding health recovery efforts for these patients. Given the heterogeneous nature of complaints and symptoms, a skilled team's individualized direction of multi-professional interventions is indispensable for recovering post-COVID-19 patients. Prolonged interventions exceeding 16 weeks are recommended to improve various biopsychosocial aspects of these patients. Furthermore, periodic evaluations can more accurately guide rehabilitation programs for post-COVID patients. A significant portion of patients with post-COVID syndrome continue to experience physical and psychological limitations that hinder their ability to perform daily and occupational activities. Therefore, promoting multidisciplinary interventions is imperative and urgent for a substantial population affected by COVID-19.

Introduction

I suggest incorporating recent publications to strengthen the context and relevance of your study. This will help highlight current trends and gaps in the literature.

Re: We inserted pieces of information about this content: Earlier studies have explored the function of physical exercise as a possible approach to mitigate the harmful impacts of COVID-19. However, they have not yet considered the various multi-professional elements related to public health promotion strategies and the severity of COVID-19 in different symptomatology (mild, moderate, severe/critical cases) [5,13,14].

Materials and Methods

The methods section shows noticeable improvement compared to the first version; congratulations on this enhancement.

Re: Thank you for your feedback.

Results & Discussion

I am satisfied with the presentation of the study results and the discussion section. The authors have effectively addressed my previous comments, resulting in a notable improvement in the overall quality of the manuscript.

Re: Thank you.

Reviewer 3 Report (New Reviewer)

Comments and Suggestions for Authors

Dear Authors,

The work you have presented is highly relevant for advancing our understanding of the recovery of COVID-19 patients. The inclusion of three groups with varying severity levels makes your study even more significant. The positive results obtained across all three groups are very encouraging and indicate that the treatment protocols were well-defined. I extend my congratulations to you.

I have a few observations:

- In the Abstract: It would be better to present the effect size numerically rather than textually.

- If you wish to group the components of the different measurements, it would also be understandable, allowing you to present some more numerical results.

- Figure 1 is not legible; I recommend improving its quality and removing the colors (perhaps you can find a better visual representation).

- Page 4: "Moreover, all evaluators have experience carrying out measurements and have already participated in other studies measuring the same indicators presented in the present study, with an intra-evaluator reproducibility greater than 0.95": The corresponding citation for this statement is missing.

- The quality of Figure 2 is also not good.

- The information about the instrument used to measure the MIHS is repeated (page 7, lines 241-243).

- The topics on "Psychoeducation protocol" (pages 8-9) could perhaps be presented in a table or another format that is easier to read.

- In the "Physical exercise protocol" (page 9), you indicate that "The training protocol consisted of cardiorespiratory and muscle strength exercises (concurrent training) to increase muscle strength and, if necessary, motor coordination and balance." How do you work on these components (motor coordination and balance), and based on what criteria do you deem it necessary to work on them or not?

Author Response

Reviewer 3:

Dear Authors,

The work you have presented is highly relevant for advancing our understanding of the recovery of COVID-19 patients. The inclusion of three groups with varying severity levels makes your study even more significant. The positive results obtained across all three groups are very encouraging and indicate that the treatment protocols were well-defined. I extend my congratulations to you.

I have a few observations:

- In the Abstract: It would be better to present the effect size numerically rather than textually.

Re: Thank you for your appointment. We excluded it.

- If you wish to group the components of the different measurements, it would also be understandable, allowing you to present some more numerical results.

Re: I am sorry, but I did not understand your question or point. Could you be more specific? Thank you.

- Figure 1 is not legible; I recommend improving its quality and removing the colors (perhaps you can find a better visual representation).

Re: We adjusted it.

- Page 4: "Moreover, all evaluators have experience carrying out measurements and have already participated in other studies measuring the same indicators presented in the present study, with an intra-evaluator reproducibility greater than 0.95.": The corresponding citation for this statement is missing.

Re: We inserted some information about this content: Moreover, all evaluators have experience carrying out measurements and have already participated in other studies measuring the same indicators presented in the present study, with an intra-evaluator reproducibility greater than 0.95 with data from our research laboratory.

- The quality of Figure 2 is also not good.

Re: We adjusted it. Thank you.

- The information about the instrument used to measure the MIHS is repeated (page 7, lines 241-243).

Re: There are two tests: maximum isometric handgrip strength and maximum isometric lumbar traction strength.

- The topics on "Psychoeducation protocol" (pages 8-9) could perhaps be presented in a table or another format that is easier to read.

Re: We changed psychoeducation and nutrition topis as a table.

- In the "Physical exercise protocol" (page 9), you indicate that "The training protocol consisted of cardiorespiratory and muscle strength exercises (concurrent training) to increase muscle strength and, if necessary, motor coordination and balance." How do you work on these components (motor coordination and balance), and based on what criteria do you deem it necessary to work on them or not?

Re: We inserted more information about it. Thank you for your feedback. Consequently, the exercise sessions were designed with low volume and frequency and performed twice weekly. This approach has shown therapeutic benefits across various chronic diseases, as evidenced by Pedersen and Saltin's review [35]. It has been effective in improving symptoms, functional capacity, and quality of life in patients with psychiatric, neurological, metabolic, cardiovascular, pulmonary, and musculoskeletal disorders and cancer. In addition, ten percent of severe/critical COVID-19 patients had their warm-up routines adapted to include coordination and balance activities. The activities performed were standing on one leg, swinging the legs, raising the legs backward, raising the arms, standing on tiptoes, walking in a straight line, sitting down and getting up from a chair, jumping over objects, using a Pilates ball, walking on a slackline, and standing on a Bosu ball – after that, they performed the physical exercises conforming table 3. Table 3 presents the training program performed by the experimental groups during the 16 weeks of multi-professional intervention.

This manuscript is a resubmission of an earlier submission. The following is a list of the peer review reports and author responses from that submission.

Round 1

Reviewer 1 Report

Comments and Suggestions for Authors

Dear all,

Thank you for the opportunity to review this manuscript. The manuscript aligns with the goal of Healthcare, and the topic offers information for researchers, professionals, and the Health Assessments section. The authors present a well-articulated study investigating multi-professional intervention effects on improving health-related physical fitness (body composition, physical fitness) and biochemical markers parameters in overweight COVID-19 survivors. I believe the information provided might need to be clarified. Consequently, some points are listed below:

Abstract

Line 25: ‘This non-randomized clinical trial included 59 (48,8% of female)’: please, could you explain to readers how you calculated this ratio? Please check.

Lines 29-36: The results provided in the abstract do not make sense. It would be constructive to incorporate detailed numerical values. Please transfer the p values and effect sizes from the results section for the reported findings.

1. Introduction

In the introduction, could you discuss the association between overweight and COVID-19 symptoms?

2. Materials and Methods

Lines 102, 176, 344: ‘in the university facilities’, ‘of the university’, ‘at the university facilities’: which university? Please, indicate more details. The location of the study should be stated.

The sample size appears to be missing. This section should be included.

The key weakness of this study is that no normal-weight participants were included to investigate these same relationships in a "healthy" comparison group. Why didn't this study involve normal-weight adults? Please add to the limitation.

Lines 121-126: doing 10 measures in one day (the 2nd day), 6 of them are physical tests [flexibility test (sit and reach test); maximal isometric strength; sit-up test; 30-s chair-stand test; push-up (adapted); and 6-min walk test (6MWT)]. for overweight and obese COVID-19 survivors? You don’t think this didn’t affect negatively the readings? How could you detect that?

Lines 154-156: You classified the 3 groups based on the symptoms of COVID-19 (mild, moderate, and severe); could you describe the symptoms variance of the three classifications?

Why didn’t you use a control group?

Figures 1&2. The resolution is not adequate; figures need optimizing display size.

Line 200: ‘Researchers respected a rest between the tests’; Were the rest intervals (recovery) fixed for all patients? Are 19 and 65 years are same? How long is the rest interval?

Providing pictures for the physical test will be more valuable/informative.

Line 284: Psychoeducation protocol; could you explain if the psychoeducational intervention was adapted based on the characteristics of the participants? (age, civil status, and education).  

Were the talks (psychoeducational intervention) concerning principles and material from psychology conducted collaboratively or separately?

3. Results

In Table 3: column 2 ‘Before’, please change to be pre-test or baseline as you mentioned in the manuscript ‘e.g., line 114’.

I wish the authors could present the differences between males and females in anthropometry and body composition, physical and cardiorespiratory fitness, and biochemical parameters after the intervention.

Discussion

Could the authors discuss the clinical characteristics of patients of the three COVID-19 survivors’ groups at baseline?

Please, can you discuss the finding without repeating words, ‘the words increased and reduction repeated 5 times each’? For example, lines 474-485, increased values for A, B, and C parameters; reduction in E, F, and G parameters).  

References

No comment

Best wishes,

Author Response

Reviewer 1

  1. Dear all, thank you for the opportunity to review this manuscript. The manuscript aligns with the goal of Healthcare, and the topic offers information for researchers, professionals, and the Health Assessments section. The authors present a well-articulated study investigating the effects of multi-professional intervention on improving health-related physical fitness (body composition, physical fitness) and biochemical markers in overweight COVID-19 survivors. I believe the information provided might need to be clarified. Consequently, some points are listed below:

RE: Thank you for your suggestions. We have adjusted all the necessary points.

Abstract

  1. Line 25: ‘This non-randomized clinical trial included 59 (48,8% of female)’: please, could you explain to readers how you calculated this ratio? Please check.

RE: The percentage refers to the number of women was calculated, taking into account the total number of participants, but this information was removed from the summary for better structuring following the recommendations of the reviews.

  1. Lines 29-36: The results provided in the abstract do not make sense. It would be constructive to incorporate detailed numerical values. Please transfer the p values and effect sizes from the results section for the reported findings.

RE: Thank you for your suggestion. We adjusted this point. The abstract has been reformulated:

“Given the multifactorial nature of COVID-19, multi-professional interventions can be an effective strategy for improving health-related indicators. Thus, this study aimed to investigate the effects of such an intervention on body composition, physical fitness, and biomarkers in overweight COVID-19 survivors with varying degrees of symptom severity after 8 weeks and 16 weeks. This non-randomized clinical trial included 59 overweight COVID-19 survivors (32 males and 27 females) divided into three groups: mild (n=31), moderate (n=13), and severe/critical (n=15). Participants underwent a multi-professional program and were assessed for anthropometric, body composition, physical fitness, and biochemical markers at 8 and 16 weeks before the intervention. After 8 weeks, time effects were observed for maximum isometric handgrip strength (p<0.001; large effect), maximum isometric lumbar-traction strength (p= 0.01; moderate effect), flexibility (p<0.001; large effect), abdominal strength-endurance (p<0.001; large effect), sit-and-stand test (p<0.001; large effect), maximum oxygen consumption (p<0.001; large effect), and distance covered in the 6-minute walk test (p<0.001; large effect). Additionally, time effects were also observed for fat mass (p= 0.03; moderate effect), body fat percentage (p= 0.02; large effect), abdominal circumference (p = 0.01; moderate effect), total cholesterol (p < 0.001; large effect), low-density lipoproteins (p < 0.001; large effect), and glycated hemoglobin (p < 0.001; large effect), with lower values after multi-professional interventions. After 16 weeks, systolic and diastolic blood pressure showed significant reductions independently of the intervention group (p<0.001; large effect). These findings suggest that multi-professional interventions can provide substantial benefits for post-COVID-19 patients, regardless of the severity of their initial symptoms.”

Introduction

  1. In the introduction, could you discuss the association between overweight and COVID-19 symptoms?

RE: Thank you for your note. We performed adjustments:

“Scientific literature showed a higher prevalence of hospitalization in obese patients with COVID-19. Rottoli et al. [11] carried out a study in a hospital in Bologna, Italy; among obese individuals, 51.9% had respiratory failure, 36.4% were admitted to the intensive care unit, 25% required mechanical ventilation, and 29.8% died within 30 days of the onset of symptoms. Gao et al. [12] investigated the association between obesity and the severity of COVID-19 in three Chinese hospitals, and found that obese individuals had more extended hospital stays and progressed to the severe form of the disease.”

  1. Materials and Methods
  2. Lines 102, 176, 344: ‘in the university facilities’, ‘of the university’, ’at the university facilities’: which university? Please, indicate more details. The location of the study should be stated.

RE: Thank you for your comment. The location of the study (Cesumar University) has been indicated.

  1. The sample size appears to be missing. This section should be included.

RE: The sample size is on line 147.

Following Jensen et al. [15], a minimum of 15 participants per group would be sufficient to achieve a statistical power of 80% with an alpha error of 5%.

  1. The key weakness of this study is that no normal-weight participants were included to investigate these same relationships in a “healthy” comparison group. Why didn’t this study involve normal-weight adults? Please add to the limitation.

RE: This work is linked to the Interdisciplinary Health Promotion Laboratory (LIIPS) at Cesumar University. The projects carried out at the LIIPS focus on overweight and obese individuals.

We inserted this limitation:

“Include a normal-weight participant to investigate these same relationships in a “healthy” comparison group.”

  1. Lines 121-126: doing 10 measures in one day (the 2nd day), 6 of them are physical tests [flexibility test (sit and reach test); maximal isometric strength; sit-up test; 30-s chair-stand test; push-up (adapted); and 6-min walk test (6MWT)]. for overweight and obese COVID-19 survivors? You don’t think this didn’t affect negatively the readings? How could you detect that?

RE: Thank you for your note. We added this information: The physical fitness tests were performed following the recommendations published earlier by Sordi et al. [5], in a clinical trial with post-COVID-19 patients. The study in question was conducted by our research group earlier the conduction of the present study.

  1. Sordi, A.F.; Lemos, M.M.; De Souza Marques, D.C.; Ryal, J.J.; Priscila de Paula Silva Lalucci, M.; Marques, M.G.; Amaro Camilo, M.L.; De Paula Ramos, S.; Franzói De Moraes, S.M.; Valdés-Badilla, P.; et al. Effects of a multi-professional intervention on body composition, physical fitness, and biochemical markers in overweight COVID-19 survivors: a clinical trial. Front Physiol. 2023, 14, 1219252.
  2. Lines 154-156: You classified the 3 groups based on the symptoms of COVID-19 (mild, moderate, and severe); could you describe the symptoms variance of the three classifications?

RE: Thank you for your suggestion. We adjusted this point.

“The clinical manifestations of COVID-19 are divided according to their severity: mild (no evidence of viral pneumonia or hypoxia); moderate (clinical signs of pneumonia, but no signs of severe pneumonia); severe (clinical signs of pneumonia plus one of the following: respiratory rate >30 breaths/min, severe respiratory distress, or SatO2 <90%) and critical (acute respiratory distress syndrome, sepsis, septic shock or acute thrombosis) [11].”

  1. Why didn’t you use a control group?

RE: While Randomized Clinical Trials (RCTs) are often regarded as the gold standard for investigating dose-response relationships and causal effects in exercise science, there are compelling reasons to omit control groups in specific contexts. One significant challenge is the difficulty in forming a control group, mainly because many individuals are asymptomatic. In Brazil’s current phase of the COVID-19 pandemic, recruiting a control group poses additional problems. Providing care for the control group is complicated by the risk of SARS-CoV-2 infection at any time. Furthermore, withholding assistance from COVID-19 survivors would be unethical, potentially exacerbating the biopsychosocial health issues faced by these individuals.

Moreover, excluding a control group can simplify the study design, making it easier to implement and analyze. This simplification is particularly advantageous when logistical constraints render the inclusion of a control group impractical. Simplified designs can also enhance the feasibility of large-scale studies by reducing the complexity associated with managing multiple groups. In conclusion, although RCTs remain a cornerstone of research in exercise science, there are solid arguments for conducting studies without control groups in specific situations. Simplified designs, cost and resource efficiency, ethical considerations, and advanced methodologies provide robust justification for this approach. Such studies have the potential to yield valuable insights and significantly advance the field of exercise science.

  1. Figures 1&2. The resolution is not adequate; figures need optimizing display size.

RE: Thank you for your note. We performed the adjustments to the figures.

  1. Line 200: ‘Researchers respected a rest between the tests’; Were the rest intervals (recovery) fixed for all patients? Are 19 and 65 years are same? How long is the rest interval?

RE: We added more information about this aspect:

“The participants were instructed about the procedures for all physical tests, and the researchers respected the remaining 5-10 min between the tests (in each test, the participants were familiarized with performing the tests).”

  1. Providing pictures for the physical test will be more valuable/informative.

RE: We would like to include photos of the participants performing the tests, but unfortunately, we do not have permission to do so. Thank you for your suggestion.

  1. Line 284: Psychoeducation protocol; could you explain if the psychoeducational intervention was adapted based on the characteristics of the participants? (age, civil status, and education).

RE: The psychoeducational interventions were adapted to the participants’ reality according to the feedback provided during the interventions. This has been adjusted in the text.

  1. Were the talks (psychoeducational intervention) concerning principles and material from psychology conducted collaboratively or separately?

RE: The meetings took place in a group once a week, for approximately 45 minutes, for 16 consecutive weeks. This has been adjusted in the text.

  1. Results

In Table 3: column 2 ‘Before’, please change to be pre-test or baseline as you mentioned in the manuscript ‘e.g., line 114’.

RE: Thank you for your note. We performed the adjustments to the tables.

  1. I wish the authors could present the differences between males and females in anthropometry and body composition, physical and cardiorespiratory fitness, and biochemical parameters after the intervention.

RE: The authors chose not to present the differences between men and women regarding the parameters studied (anthropometry and body composition, physical and cardiorespiratory fitness, and biochemical parameters) due to the low number of participants. Therefore, considering that our sample was reduced at the end of the analyses, we would not have sufficient power to carry out analyses by sex.

Discussion

  1. Could the authors discuss the clinical characteristics of patients of the three COVID-19 survivors’ groups at baseline?

RE: In major variables, we did not identify significant differences. However, we insert in the discussion section some points about the differences detected:

“A previous study reported that patients with a severe/critical form of COVID-19 showed higher values of fat mass and body fat percentage than those with a mild form of COVID-19 with the same BMI [7]. Fat mass was significantly higher in severe/critical COVID-19 compared to the mild group; similar data were found by Perli et al. [6] when comparing the different symptoms of the disease; the authors also observed that this difference persisted 1 year after the disease”.

  1. Perli, V.A.S.; Sordi, A.F.; Lemos, M.M.; Fernandes, J.S.A.; Capucho, V.B.N.; Silva, B.F.; De Paula Ramos, S.; Valdés-Badilla, P.; Mota, J.; Branco, B.H.M. Body composition and cardiorespiratory fitness of overweight COVID-19 survivors in different se-verity degrees: a cohort study. Sci Rep. 2023, 13, 17615.
  2. Please, can you discuss the finding without repeating words, ‘the words increased and reduction repeated 5 times each’? For example, lines 474-485, increased values for A, B, and C parameters; reduction in E, F, and G parameters).

RE: Thank you for the observation.

The following outcomes were observed: (i) 8 weeks of intervention showed improvements in MIHS values for the right and left sides, MILTS, push-ups, abdominal strength endurance repetitions, sit and stand test, VO2peak and distance covered in the 6MWT; (ii) increased flexibility values after 8 and 16 weeks; (iii) pre-test and final DBP showed a significant reduction after 16 weeks and abdominal circumference after 8 weeks; (iv) the severe group showed an increase in fat mass values and body fat percentage compared to the mild group; (v) 8 weeks showed improvements in total cholesterol, LDL-c, HbA1c, urea and in GGT, lipase and magnesium reduction values after 16 weeks; (vi) the severe group showed an increase in creatinine values compared to the moderate group; (vii) CRP had a significant increase in the severe/critical group compared to the mild group; and (viii) it was observed that the 8- and 16-week interventions showed significant improvements in body composition parameters, physical fitness and biomarkers.

  1. References

No comment

RE: Thank you for your feedback.

Reviewer 2 Report

Comments and Suggestions for Authors

First of all, the reviewer would like to thank the authors for their work and efforts in trying to improve  heatlth science  healthknowledge.

The aim of the papper was to investigate the effects of such an intervention on body composition, physical fitness, and biomarkers in overweight COVID-19 survivors with varying degrees of symptom severity (mild, moderate, and severe/critical) after 8 and 16 weeks.

I consider it an interesting paper but it needs some modifications to be taken into consideration for publication.

Methods:

- Data collection should indicate the personnel who performed the data and the experience they had.

- Line 150. I find it strange that out of 146 volunteers only 75 met the inclusion criteria. It should be indicated which criteria were not met and to what extent. Did all the volunteers have a BMI>25? How was the invitation to participate made, the BMI calculation is made once the intervention is started. This point should be further clarified. It is common for the BMI of these patients to be high, but I find it curious that all the volunteers met it.

- Postcovid patients have a very high difficulty to exercise load, intensity or any stimulus. How was this in your study, how was it controlled? Were exercises adapted according to symptomatology?

- Also, I am surprised at the degree of adherence to the program.  Did those you counted as a sample complete the entire program with all the sessions?  This fact surprises me very much given the instability of the symptomatology of these patients.

- I think the authors should detail what health problems or complications were determinant for a participant to be a loss.

- Line 215. I understand that the isometric strength assessed is manual grip strength. It should be stated.

- Line 221: What data were taken from the 6MWT? Only the distance traveled or also heart rate and Borg scale?

- Why was the exercise program only performed on two days when most clinical guidelines indicate a minimum frequency of 3 days per week of cardiorespiratory exercise and a minimum of 2 days per week of strength work?

- It seems very appropriate to me that the exercise physiologist had a lot of experience in chronic pathology. But the exercise protocol should be supported by scientific evidence. I ask the authors to make the effort to explain, based on evidence, why the parameters used. In addition to indicating the intensities and loads at which both muscle strength trainings and aerobic exercise were performed.

- How were the rest times between exercises and sets managed?

- What did the warm-up consist of?

Results

- The results should be reformulated. The information in the tables should not be duplicated in the text. It would be advisable to make the text more explanatory and leave the values in the tables.

Author Response

Reviewer 2:

  1. First of all, the reviewer would like to thank the authors for their work and efforts in improving health science knowledge. The paper aimed to investigate the effects of such an intervention on body composition, physical fitness, and biomarkers in overweight COVID-19 survivors with varying degrees of symptom severity (mild, moderate, and severe/critical) after 8 and 16 weeks. I consider it an interesting paper, but some modifications must be considered before it can be published.

RE: Thank you for your suggestions. We have adjusted all the necessary points.

Methods:

  1. Data collection should indicate the personnel who performed the data and their experience.

RE: We inserted information about this content:

“All evaluators have experience carrying out measurements and have already participated in other studies measuring the same indicators presented in the present study, with an intra-evaluator reproducibility greater than 0.95.”

  1. Line 150. I find it strange that out of 146 volunteers, only 75 met the inclusion criteria. It should be indicated which criteria were not met and to what extent. Did all the volunteers have a BMI>25? How was the invitation to participate made, the BMI calculation is made once the intervention is started. This point should be further clarified. It is common for the BMI of these patients to be high, but I find it curious that all the volunteers met it.

RE: Out of the 71 individuals excluded, a significant number of volunteers were not granted medical clearance to engage in physical exercise due to myocarditis or other cardiac conditions, which prevented their participation in the study. As a result, there was a notable discrepancy between the eligible participants and those excluded before the study commenced. Participant registration was conducted via a link where they provided basic information such as age, body weight, height, and other health conditions.

  1. Postcovid patients have a very high difficulty with exercise load, intensity, or any stimulus. How was this in your study, how was it controlled? Were exercises adapted according to symptomatology?

RE: We insert a text about this content:

“The rest interval between exercises varied for each participant, depending on their physical condition. This interval was individualized based on each participant’s readiness to perform the exercise.”

  1. Also, I am surprised at the degree of adherence to the program. Did those you counted as a sample complete the entire program with all the sessions? This fact surprises me very much given the instability of the symptomatology of these patients.

RE: They contracted COVID-19 between January 2, 2021, and September 22, 2021. Our study was conducted from January to October 2022. Only the more stable patients were included in the study, while unstable participants were not accepted.

  1. I think the authors should have detailed what health problems or complications were determinants of a participant’s loss.

RE: Ok, we insert this point:

“Another point that needs to be considered is dropouts (health problems; n = 9) from the present study related to low %SpO2 during physical exercise and at rest and medical advice for temporarily suspending practical activities.”

  1. Line 215. I understand that the isometric strength assessed is manual grip strength. It should be stated.

RE: It was changed.

  1. Line 221: What data were taken from the 6MWT? Only the distance traveled or also heart rate and Borg scale?

RE: We collected distance, heart rate, systolic and diastolic pressure (before) and after 6MWT, conforming to Table 4.

  1. Why was the exercise program only performed on two days when most clinical guidelines indicate a minimum frequency of 3 days per week of cardiorespiratory exercise and a minimum of 2 days per week of strength work?

RE: Due to restrictions from the COVID-19 pandemic and/or risk of infections, it was decided to carry out physical activities only twice a week to reduce the exposure of participants and study staff to possible risks of contracting SARS-CoV-2.

It seems very appropriate to me that the exercise physiologist had a lot of experience in chronic pathology. But the exercise protocol should be supported by scientific evidence. I ask the authors to try to explain, based on evidence, why the parameters were used. In addition, the intensities and loads at which both muscle strength training and aerobic exercise were performed were indicated.

RE: We inserted pieces of information about this content.

“The physical exercise intervention sessions were held at the Cesumar University facilities twice weekly, lasting 60 minutes each, with training alternating programs A and B on different days. The exercise physiologists developed the study protocol specialists based on their chronic disease experience and responses to health-related physical fitness tests. The training protocol consisted of cardiorespiratory and muscle strength exercises (concurrent training) to increase muscle strength and, if necessary, motor coordination and balance. The concurrent training plan consisted of 4 weeks of anatomical adaptation with low volume and intensity, that is, 3 sets of 15 repetitions, with no aerobic exercise at the end of the session, and the remaining weeks of physical exercise (12 more in total) had a gradual progression of volume and intensity (via the classic linear periodization); in other words, the loads used were readjusted over the weeks, as well as the number of sets and repetitions. In weeks 5 to 8, 3 sets of 12 repetitions were performed; in weeks 9 to 12, the training sessions consisted of 4 sets of 12 repetitions; and finally, in weeks 13 to 16, 4 sets of 20 repetitions were performed. This protocol was based on/adapted from the interventions with post-COVID-19 patients conducted earlier by Sordi et al. [5]. About aerobic exercise, 1 set of 5 and 1 set of 10 minutes was performed on weeks 5 to 8; 1 set of 10 and 1 set of 15 minutes was performed on weeks 9 to 12; and finally, on weeks 13 to 16, 1 set of 10 and 1 set of 20 minutes was performed. The training was carried out with resistance exercises focused on large muscle groups and cardiorespiratory fitness, which were performed on a treadmill, vertical/horizontal bicycle, or rowing ergometer, according to the preference and physical condition of the patients. The rest interval between exercises varied for each participant, depending on their physical condition. This interval was individualized based on each participant’s readiness to perform the exercise. For a third day, the participants were requested to improve their physical activity (especially walking, 1 hour per week - if possible, accompanying the physical training).”

  1. Sordi, A.F.; Lemos, M.M.; De Souza Marques, D.C.; Ryal, J.J.; Priscila de Paula Silva Lalucci, M.; Marques, M.G.; Amaro Camilo, M.L.; De Paula Ramos, S.; Franzói De Moraes, S.M.; Valdés-Badilla, P.; et al. Effects of a multi-professional intervention on body composition, physical fitness, and biochemical markers in overweight COVID-19 survivors: a clinical trial. Front Physiol. 2023, 14, 1219252.
  2. How were the rest times between exercises and sets managed?

RE: This part has been added:

“The rest interval between exercises varied for each participant, depending on their physical condition. This interval was individualized based on each participant’s readiness to perform the exercise.”

  1. What did the warm-up consist of?

RE: We inserted this information in Table 1. Warm-up (walk, cycle, or rowing at moderate intensity for 8 min).

  1. Results

- The results should be reformulated. The information in the tables should not be duplicated in the text. Making the text more explanatory and leaving the values in the tables would be advisable.

RE: Thank you for your comment. To avoid duplication of results, we have chosen to report the results of each variable (e.g., physical tests, biochemical markers) in tables and to present only the results of the statistical analyses as text.

Reviewer 3 Report

Comments and Suggestions for Authors

Dear Authors,

First and foremost, I would like to commend you for your efforts and the clarity with which the study has been presented. The topic is indeed compelling; however, the paper in its current form is not yet suitable for publication. Nonetheless, I believe there is substantial merit in the research approach, and I encourage you to address the following points to enhance the manuscript:

Abstract

  • Line 25: Please include the percentage of male participants.
  • Line 29: Specify whether the time points are at 8 and 16 weeks, respectively.

Introduction

  • The introduction lacks sufficient detail on multi-professional interventions and their effects on individuals with a history of COVID-19. Please expand on this aspect.
  • Provide a clearer differentiation of symptoms (e.g., mild, moderate, severe/critical) as they relate to the study.
  • Line 88: Consider rephrasing “superior effectiveness” for better clarity.

Materials and Methods

  • Line 134: Figure 2 should be referred to as Figure 1. Please verify and correct the figure numbering as necessary.
  • Line 158: Figure 1 appears to be misplaced. It should be moved to the end of Section 2.1, as it pertains to the “experimental approach to the problem.”
  • Figure 1 needs to be presented in higher resolution.
  • The subtitles following “Health-related physical fitness tests” need correction. Either renumber these subtitles to align with the 4th-degree title (2.5.1.1) or remove the subtitle numbers altogether, as they all relate to Section 2.5.1.

Results

  • The results and tables are satisfactory and accurately reflect the measured variables and findings.

Discussion

  • The discussion section is overly lengthy, which may hinder understanding, especially for readers not directly familiar with the topic. I recommend condensing this section to focus more sharply on the main findings and their implications.
  • Lines 486-496: Please provide interpretations for the insignificant findings reported in this range.
  • Lines 512-529: Further elucidation is needed on the general differences among mild, moderate, and severe/critical COVID-19 groups, and how your study contributes to this understanding.
  • Line 342: This part appears subjective and may impact the overall findings. Consider revising it for greater objectivity.

Conclusions

  • The conclusions section is also excessively detailed, resembling a thesis in length. Please revise it to highlight the main and most crucial findings of the study.
  • Update any outdated references to ensure the most current and relevant sources are cited.
Comments on the Quality of English Language

Minor editing of English language required.

Author Response

  1. First and foremost, I would like to commend you for your efforts and the clarity with which the study has been presented. The topic is indeed compelling; however, the paper in its current form is not yet suitable for publication. Nonetheless, I believe there is substantial merit in the research approach, and I encourage you to address the following points to enhance the manuscript.

RE: Thank you for your feedback. We adjusted all the points, conforming as requested.

Abstract

  1. Line 25: Please include the percentage of male participants.

RE: It was inserted:

“This non-randomized clinical trial included 59 overweight COVID-19 survivors (32 males and 27 females) divided into three groups: mild (n=31), moderate (n=13), and severe/critical (n=15).”

  1. Line 29: Specify whether the time points are at 8 and 16 weeks, respectively.

RE: It has been changed:

“Participants underwent a multi-professional program and were assessed for anthropometric, body composition, physical fitness, and biochemical markers at 8 and 16 weeks before the intervention.”

Introduction

  1. The introduction lacks sufficient detail on multi-professional interventions and their effects on individuals with a history of COVID-19. Please expand on this aspect.

RE: We added this point in the introduction section:

“Scientific literature showed a higher prevalence of hospitalization in obese patients with COVID-19. Rottoli et al. [11] carried out a study in a hospital in Bologna, Italy; among obese individuals, 51.9% had respiratory failure, 36.4% were admitted to the intensive care unit, 25% required mechanical ventilation, and 29.8% died within 30 days of the onset of symptoms. Gao et al. [12] investigated the association between obesity and the severity of COVID-19 in three Chinese hospitals, and found that obese individuals had more extended hospital stays and progressed to the severe form of the disease.”

  1. Rottoli, M.; Bernante, P.; Belvedere, A.; Balsamo, F.; Garelli, S.; Giannella, M.; Cascavilla, A.; Tedeschi, S.; Ianniruberto, S.; Rosselli Del Turco, E,; et al. How important is obesity as a risk factor for respiratory failure, intensive care admission and death in hospitalised COVID-19 patients? Results from a single Italian centre. Eur J Endocrinol. 2020, 183, 389-397.
  2. Gao, F.; Zheng, K.I.; Wang, X.B.; Sun, QF; Pan, K.H.; Wang, T.Y; Chen, Y.P.; Targher, G.; Byrne, C.D.; George, J., et al. Obesity Is a Risk Factor for Greater COVID-19 Severity. Diabetes Care. 2020, 43, e72-4.
  3. Provide a clearer differentiation of symptoms (e.g., mild, moderate, severe/critical) as they relate to the study.

RE: We added this point:

“The clinical manifestations of COVID-19 are divided according to their severity: mild (no evidence of viral pneumonia or hypoxia); moderate (clinical signs of pneumonia, but no signs of severe pneumonia); severe (clinical signs of pneumonia plus one of the following: respiratory rate >30 breaths/min, severe respiratory distress, or SatO2 <90%) and critical (acute respiratory distress syndrome, sepsis, septic shock or acute thrombosis) [13].”

  1. World Health Organization (WHO). COVID-19 clinical management: living guidance. WHO, 2023; pp. 1-116.
  2. Line 88: Consider rephrasing “superior effectiveness” for better clarity.

RE: We changed to: “better results…”

Materials and Methods

  1. Line 134: Figure 2 should be referred to as Figure 1. Please verify and correct the figure numbering as necessary.

RE: We changed the point of Figure 2. Thank you.

  1. Line 158: Figure 1 appears to be misplaced. It should be moved to the end of Section 2.1, as it pertains to the “experimental approach to the problem.”

RE: It has been changed.

  1. Figure 1 needs to be presented in higher resolution.

RE: We adjusted it. Thank you.

  1. The subtitles following “Health-related physical fitness tests” need correction. Either renumber these subtitles to align with the 4th-degree title (2.5.1.1) or remove the subtitle numbers altogether, as they all relate to Section 2.5.1.

RE: We adjusted it. Thank you.

Results

  1. The results and tables are satisfactory and accurately reflect the measured variables and findings.

RE: Thank you.

Discussion

  1. The discussion section is overly lengthy, which may hinder understanding, especially for readers not directly familiar with the topic. I recommend condensing this section to focus more sharply on the main findings and their implications.

RE: We tried to reduce some content in the discussion section.

  1. Lines 486-496: Please provide interpretations for the insignificant findings reported in this range.

RE: We discussed it: However, the patient’s biochemical analyses were among normative values at pre-intervention [40].

  1. Williamson, M.A.; Synder, L.M. Wallach’s interpretation of diagnostic tests (Portuguese). 10º ed.; Guanabara Koogan: Rio de Janeiro, Brasil, 2015; pp. 1-1288.
  2. Lines 512-529: Further elucidation is needed on the general differences among mild, moderate, and severe/critical COVID-19 groups, and how your study contributes to this understanding.

RE: We added some information about it:

“The clinical manifestations of COVID-19 are divided according to their severity: mild (no evidence of viral pneumonia or hypoxia); moderate (clinical signs of pneumonia, but no signs of severe pneumonia); severe (clinical signs of pneumonia plus one of the following: respiratory rate >30 breaths/min, severe respiratory distress, or SatO2 <90%) and critical (acute respiratory distress syndrome, sepsis, septic shock or acute thrombosis) [13].”

  1. World Health Organization (WHO). COVID-19 clinical management: living guidance. WHO, 2023; pp. 1-116.
  2. Line 342: This part appears subjective and may impact the overall findings. Consider revising it for greater objectivity.

RE: It was adjusted. Thank you.

Conclusions

  1. The conclusions section is also excessively detailed, resembling a thesis in length. Please revise it to highlight the main and most crucial findings of the study.

RE: It was adjusted.

“The study concludes that an 8-week intervention significantly improved anthropo-metrics, body composition, physical fitness, and reductions in lipid profile and glycated hemoglobin. After 16 weeks, these variables stabilized, indicating lasting benefits. The multi-professional intervention model proved beneficial for post-COVID-19 patients, regardless of symptom severity. The study’s clinical trial design, incorporating patients with varying COVID-19 symptoms, enhances the validity and broadens the understanding of a multi-professional program’s effects. This approach combines nutritional education to promote healthy eating, psychoeducation for knowledge and behavior change, and biochemical tests to assess impacts on overweight and obese individuals.”

The severity of COVID-19 symptoms directly influences the progression of physical exercise programs, necessitating individualized training based on initial evaluations. Professionals must conduct comprehensive assessments, including physical fitness and recent blood results, to devise effective strategies for improving quality of life. Addressing individual needs and sequelae is crucial for recovering health-related physical fitness in COVID-19 survivors. Continuous monitoring and follow-up are essential and urgent. The study underscores the importance of developing strategies to improve health conditions through physical activity, nutrition, and psychoeducation, aiming for comprehensive care and support for COVID-19 survivors.

  1. Update any outdated references to ensure the most current and relevant sources are cited.

RE: We adjusted it and inserted some recent references about the theme.

Round 2

Reviewer 2 Report

Comments and Suggestions for Authors

Methods:

  1. Data collection should indicate the personnel who performed the data and their experience.

RE: We inserted information about this content:

“All evaluators have experience carrying out measurements and have already participated in other studies measuring the same indicators presented in the present study, with an intra-evaluator reproducibility greater than 0.95.”

This point is only indicated in section 2.5.1. when there are also other variables and measurements that have been taken in addition to those related to Physical fitness.

4.The rest interval between exercises varied for each participant, depending on their physical condition. This interval was individualized based on each participant’s readiness to perform the exercise.”

How did you quantify that participants were able to resume exercise? Did you use a subjective effort strategy such as the BORG scale? It is important to know how the decision to return to exercise was made.

5. RE: They contracted COVID-19 between January 2, 2021, and September 22, 2021. Our study was conducted from January to October 2022. Only the more stable patients were included in the study, while unstable participants were not accepted.

All this should be clarified in the paper. The authors aim to establish an 8-16 week program at different degrees of symptom severity.

I have many questions about this.

1. It gives the impression that the patients are soon after the acute stage of involvement. If this is not the case, it seems that patients continue to have symptomatology and therefore have developed LONG-COVID. This should be stated and how it has been diagnosed.

2.    If these are patients who no longer have symptomatology, this should also be clarified.

3.    Why did they wait one year after infection with COVID-19?

4.    Could the authors justify how they decided which patients were stable and which were not. Patients with LONG COVID, and more so those with a lot of symptomatology and chronic fatigue, tend to be very unstable in terms of vasovagal, pain, respiratory involvement. The selection of “the most stable patients” means that this study may have a very large bias if we are not able to detail how the sample was selected, since the criteria indicated by the authors are not sufficient..

  1. Why was the exercise program only performed on two days when most clinical guidelines indicate a minimum frequency of 3 days per week of cardiorespiratory exercise and a minimum of 2 days per week of strength work?

RE: Due to restrictions from the COVID-19 pandemic and/or risk of infections, it was decided to carry out physical activities only twice a week to reduce the exposure of participants and study staff to possible risks of contracting SARS-CoV-2.

However, I think that the authors should make an effort to justify why they consider that with 2 training sessions they could achieve good results.

Regarding the justification of the program, I think that the authors do not justify it properly. They only indicate that they are based on a program already carried out on an article with practically the same title as this one, with a very similar sample. On the one hand, it seems that they have segmented the information into two articles. I do not see this as a problem, since the design is different and I understand that the information analyzed is different. However, it does not seem right to me that the justification of the program is based exclusively on this evidence.

  1. What did the warm-up consist of?

RE: We inserted this information in Table 1. Warm-up (walk, cycle, or rowing at moderate intensity for 8 min).

How did you establish and control that the intensity was moderate? What parameters? VO2max? Thresholds? BORG scale? Speech test? It is crucial to indicate how the intensity was controlled both in the warm-up and in the main part.
